# Cryo-EM structure of the Rous sarcoma virus octameric cleaved synaptic complex intasome

Krishan K. Pandey [1,8 ✉], Sibes Bera[1,8], Ke Shi[2,8], Michael J. Rau [3], Amarachi V. Oleru[4], James A. J. Fitzpatrick [3,5,6], Alan N. Engelman [4,7], Hideki Aihara [2 ✉] & Duane P. Grandgenett [1 ✉]

Despite conserved catalytic integration mechanisms, retroviral intasomes composed of integrase (IN) and viral DNA possess diverse structures with variable numbers of IN subunits. To investigate intasome assembly mechanisms, we employed the Rous sarcoma virus (RSV) IN dimer that assembles a precursor tetrameric structure in transit to the mature octameric intasome. We determined the structure of RSV octameric intasome stabilized by a HIV-1 IN strand transfer inhibitor using single particle cryo-electron microscopy. The structure revealed significant flexibility of the two non-catalytic distal IN dimers along with previously unrecognized movement of the conserved intasome core, suggesting ordered conformational transitions between intermediates that may be important to capture the target DNA. Single amino acid substitutions within the IN C-terminal domain affected intasome assembly and function in vitro and infectivity of pseudotyped RSV virions. Unexpectedly, 17 C-terminal amino acids of IN were dispensable for virus infection despite regulating the transition of the tetrameric intasome to the octameric form in vitro. We speculate that this region may regulate the binding of highly flexible distal IN dimers to the intasome core to form the octameric complex. Our studies reveal key steps in the assembly of RSV intasomes.

[1] Department of Molecular Microbiology and Immunology, School of Medicine, Saint Louis University, St. Louis, MO, USA. [2] Department of Biochemistry, Molecular Biology and Biophysics, University of Minnesota, Minneapolis, MN, USA. [3] Washington University Center for Cellular Imaging, Washington University School of Medicine, St. Louis, MO, USA. [4] Department of Cancer Immunology and Virology, Dana-Farber Cancer Institute, Boston, MA, USA. [5] Departments of Cell Biology & Physiology and Neuroscience, Washington University in St. Louis, School of Medicine, St. Louis, MO, USA. [6] Department of Biomedical Engineering, Washington University in St. Louis, St. Louis, MO, USA. [7] Department of Medicine, Harvard Medical School, Boston, MA, USA. [8]These authors contributed equally: Krishan K. Pandey, Sibes Bera, Ke Shi. ✉email: krishan.pandey@health.slu.edu; aihar001@umn.edu; Duane.Grandgenett@health.slu.edu

Integration of retroviral DNA by the viral integrase (IN) into cellular DNA is a necessary step for virus replication. The structures of different retroviral IN-DNA complexes capable of concerted integration show diverse organizations but possess significant mechanistic similarities. These structures, generally termed intasomes, contain multimers of IN subunits ranging from 4 for the simiispumavirus prototype foamy virus (PFV)[1,2] and delta-retroviruses human T-cell leukemia virus type 1 (HTLV-1)[3] and simian T-lymphotropic virus type 1 (STLV-1)[4] to 8 for alpha-retrovirus Rous sarcoma virus (RSV)[5] and beta-retrovirus mouse mammary tumor virus (MMTV)[6]. For lentiviral intasomes, there are variable numbers of IN subunits ranging from 4 to 12 for HIV-1 and simian immunodeficiency virus (SIV)[7–9] and up to 16 subunits for maedi-visna virus (MVV)[10].

We previously utilized X-ray crystallography to determine the structure of the RSV strand transfer complex (STC) at 3.8 Å resolution, which revealed 4 IN dimers with the branched viral-target DNA substrate that mimicked the end-product of the concerted integration reaction[5]. RSV IN is homodimeric in solution[11–13]. Two proximal RSV IN dimers comprise the catalytic IN subunits that engage the viral DNA ends while a distal pair of non-catalytic IN dimers bridged between the two viral DNA molecules and helped capture the target DNA[5].

The assembly mechanisms of retroviral intasomes are poorly understood. Our previous studies suggested that a tetrameric RSV intasome is the precursor of the octameric intasome[14–16]. We have now determined the cryo-EM structure of the RSV octameric intasome stabilized by the HIV-1 IN strand transfer inhibitor (INSTI) MK-2048. The 3′ OH cleavage of blunt-ended viral DNA by IN yields intasomes with recessed ends, here termed the cleaved synaptic complex (CSC)[17–19]. Stabilizing interactions that occur between the INSTIs, recessed viral DNA ends and Mg++ ions within the conserved intasome core (CIC) are similar to the corresponding structures in PFV[1], HIV-1[8], and SIV[9]. In contrast to the stable association of distal IN dimers in the RSV STC[5], the two distal dimers are highly flexible in the CSC. Previous site-directed mutational studies of RSV IN[12,15,16] and the structure of RSV STC[5] suggested important roles of the IN C-terminal domain (CTD) for viral DNA binding and CSC assembly. Single-round infection assays using RSV pRIAS-Luc vectors[20] that contained wild type (wt) and IN mutants were performed to determine the biological roles of the CTD and the "tail" region of IN (C-terminal residues 269–286) in the assembly of the RSV CSC under physiological conditions.

## Results

**Structure of the RSV octameric CSC.** As we reported previously, RSV IN 1–278, which has a C-terminal truncation of 8 amino acids from the full-length 286-residue IN forms a stable octameric complex with 3′ OH recessed viral DNA substrates[15] (Supplementary Figs. 1–3). Thus, we determined the structure of the octameric CSC formed with IN 1–278 and GU3 18R viral DNA in the presence of INSTI MK-2048 by single-particle cryo-EM (Fig. 1a). The final map had an overall resolution of 3.2 Å (Supplementary Figs. 4, 5) and local resolution reaching ~2.8 Å for the core region of the complex containing the two IN catalytic sites bound with INSTIs and viral DNA (Supplementary Fig. 5), which was significantly improved from the previously reported RSV STC X-ray crystal structure that was determined at 3.8 Å resolution[5]. Well-defined cryo-EM density was observed for both of the proximal IN dimers, all four CTDs of both distal (flanking) IN dimers, two viral DNAs, and the two MK-2048 molecules occupying the active sites of the inner catalytic IN molecules (Fig. 1b, c and 2) (Supplementary Fig. 6) (Table 1). IN residues downstream from Ile269 were not discernable in the EM map (Supplementary Fig. 3).

A comparison between the refined model of the octameric RSV CSC and the crystal structure of RSV STC showed a similar overall arrangement of the IN domains and DNA molecules in

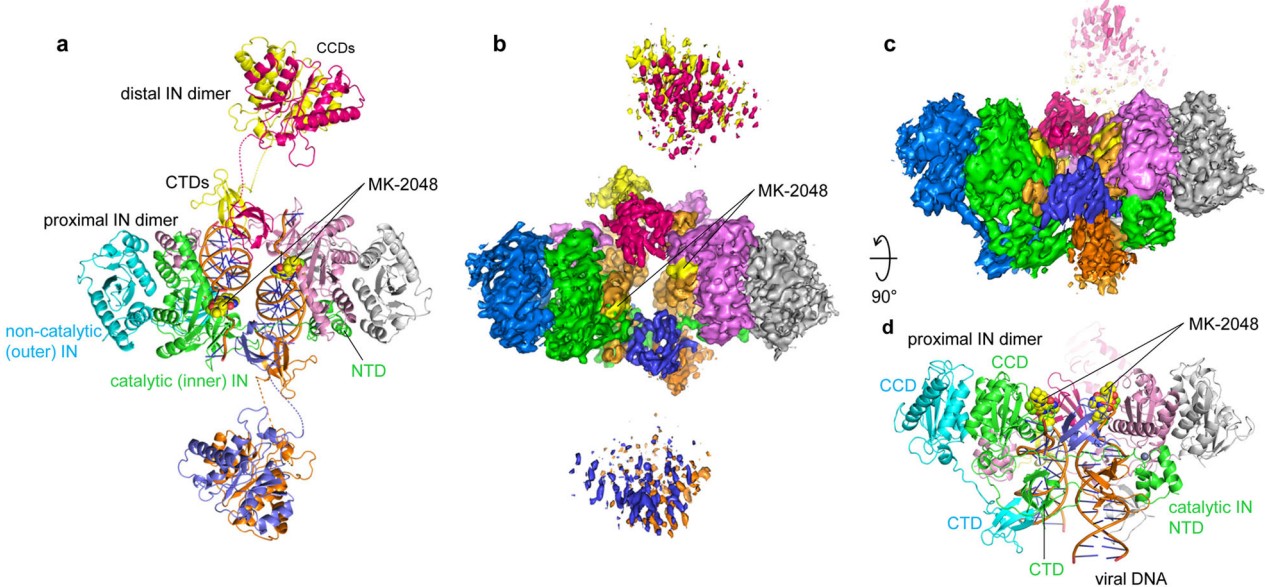

**Fig. 1 Cryo-EM structure of the RSV cleaved synaptic complex (CSC) trapped with MK-2048. a** Cartoon representation of RSV CSC viewed down its pseudo twofold axis, with the 8 IN and viral DNA molecules colored differently. The two MK-2048 molecules in the active site of the catalytic IN protomers are shown in the space-filling model. Disordered CCD-CTD linkers for the distal IN dimers are shown as dashed lines. The two proximal IN dimers (green/cyan and pink/gray) plus the distal IN CTDs (magenta, slate blue) bridging between them constitute the core of the complex, referred to as 'conserved intasome core (CIC)'. **b** Cryo-EM density in the same color scheme and orientation as in (**a**). The densities for MK-2048 are indicated and highlighted in yellow. **c** Side view after a 90-degree rotation about a horizontal axis from (**b**). The densities for distal IN CCDs in the foreground are omitted for clarity. **d** Model representation corresponding to the view in (**c**).

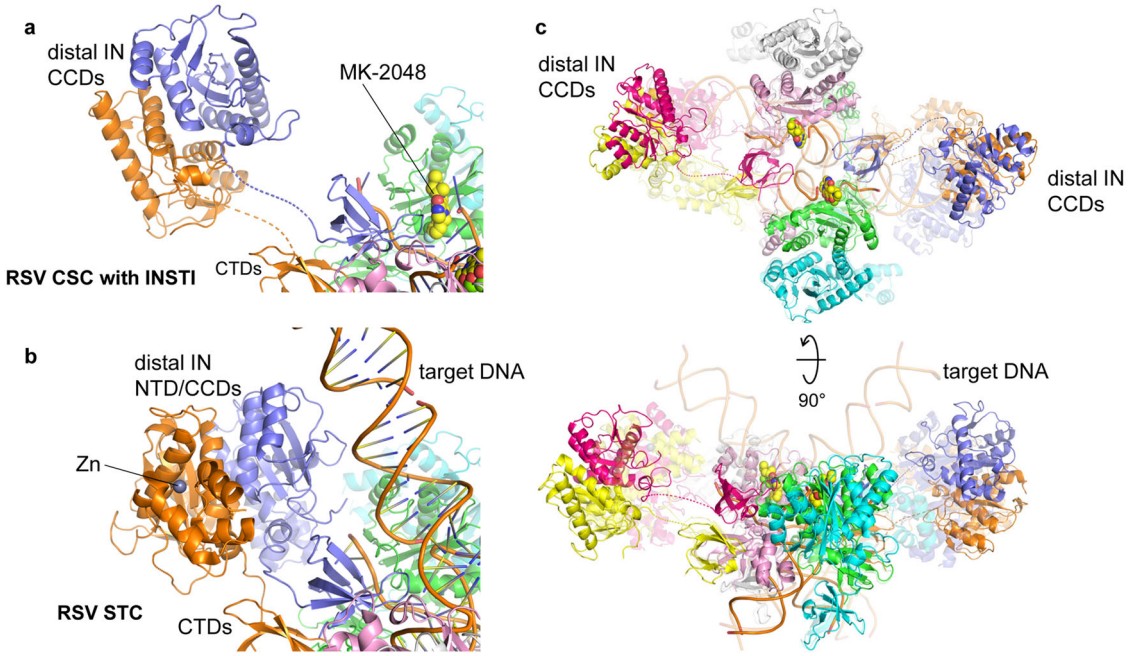

**Fig. 2 Comparison between the RSV CSC and STC structures. a** A view of the RSV CSC trapped with MK-2048, centered on one of the distal IN CCD dimers. NTD of these IN molecules was not modeled due to poor EM density, presumably reflecting high flexibility. CCD-CTD linkers of the distal IN dimer are shown as dashed lines. MK-2048 in the active site of the catalytic IN molecule (green) is shown in the space-filling model. **b** RSV STC viewed from the same angle as in (**a**). Note significant rotation and translation of the distal IN CCDs and ordering of the CCD-CTD linkers accompanying the target DNA binding (STC in comparison to CSC). The Zn-bound distal IN NTDs were modeled in the RSV STC crystal structure. **c**. Superposition of the RSV CSC (solid colors) and STC (transparent) structures in two different views. The color scheme follows that in Fig. 1a.

the CIC regions. Subtle but noticeable differences included a greater separation between the two juxtaposed proximal IN dimers in the CSC by an additional ~4 Å across the synaptic interface, which enabled the complex to "expand" slightly. Thus, the IN subunits in the CIC appears to be more loosely packed in the RSV CSC than in the STC (Fig. 2c). However, the biggest difference between the CSC and STC structures is the positioning of the NTDs and CCDs of the distal IN dimers, which were observed to bind to the distal region of the target DNA in the STC (Fig. 2a, b). The cryo-EM map of the CSC structure shows that the CCD of the distal IN dimers position farther away from the CIC region, rendering a more cruciform-like, overall open conformation of the CSC when viewed along the pseudo twofold axis. The NTDs associated with these CCDs were not modeled due to low-quality density in these regions. Weaker and less well-defined density compared to the rest of the complex suggested that the NTDs and CCDs of the distal IN dimers are highly flexible in the absence of target DNA.

**Interactions with INSTI MK-2048 in the RSV CIC.** MK-2048, the INSTI included during complex formation, is bound at the interface between IN and the viral DNA 3′ terminus, sequestering the two magnesium ions in the active site (Figs. 1 and 3a) (Supplementary Fig. 6). The chloro-fluorobenzyl moiety stacks on the penultimate cytosine base of the terminal CA dinucleotide motif, displacing the terminal deoxyadenosine (dA). The displaced dA in turn stacks on the core metal-coordinating ring system of MK-2048 (Fig. 3b). Although this mode of binding is similar to that observed for MK-2048 in the PFV intasome[21] (PDB ID: 3oyb), RSV IN makes several unique contacts with MK-2048, involving the side chain of Ser150 positioned ~3.2 Å from the aromatic ring system of MK-2048 (corresponding to Pro214 of PFV IN) and Gln151 stacked on the guanine base opposite the aforementioned cytosine and positioned ~3.3 Å from the fluorine

atom in MK-2048 (Supplementary Fig. 7a); corresponding Gln215 side chain of PFV IN is positioned farther away. Notably, Arg263 from the CTD of a distal IN subunit, which bridges between the CCD and CTD of the catalytic IN molecule as part of the CIC, interacts with the Gln151 side chain as well (Supplementary Fig. 6). A similar interaction between INSTI-Gln146-Arg263 has been observed in recent cryo-EM studies of HIV-1[8] and SIV[9] CSC intasomes.

Infection of cells by RSV is sensitive to inhibition by the first-generation INSTI raltegravir but not elvitegravir (EVG). The P145S change in HIV-1 IN confers EVG resistance, and the corresponding Ser150 residue in RSV IN conferred natural resistance to EVG[22]. The S150P change in IN increased RSV sensitivity to EVG ~74-fold[22]. The resistance to inhibition of RSV IN concerted integration by EVG was 84-fold higher than that observed for raltegravir; in contrast to raltegravir, EVG failed to trap the RSV CSC assembly in vitro[14]. Our CSC structure with natural Ser150 residue accordingly afforded the opportunity to investigate the structural basis of INSTI resistance. Ser150 interacts with MK-2048 (aromatic rings) via π-donor H-bonds (Supplementary Fig. 7a). Upon modeling, this interaction is absent with EVG as well as the π–π stacking of the fluorobenzyl moiety with the cytosine base of the catalytic DNA strand (Supplementary Fig. 7b). Further modeling of mutant S150P appears to restore the π–π stacking of EVG with the cytosine base and π-alkyl interactions of other aromatic rings of EVG with Pro150 (Supplementary Fig. 8b). The natural Pro145 in HIV-1 and SIV IN contributes hydrophobic interactions to INSTI binding[8,9]. EVG wraps around the analogous Pro214 in the PFV CSC[1], which mimics binding of EVG to the modeled S150P RSV CSC (Supplementary Fig. 8b). In addition, in our modeled structure, Gly148 of RSV IN hydrogen-bonds with the 1-hydroxy-3-methylbutan-2-yl moiety of EVG, which is a unique interaction not observed for PFV IN. The corresponding residue

**Table 1 cryo-EM data collection, refinement, and validation statistics.**

| | RSV CSC (EMDB-22400, PDB 7JN3) | Cluster E CSC (EMDB-23035, PDB 7KU7) | Cluster E CIC (PDB 7KUI) |
|---|---|---|---|
| **Data collection and processing** | | | |
| Microscope | Titan Krios | | |
| Detector | Gatan K2 | | |
| Magnification | 105,000 X | | |
| Voltage (kV) | 300 | | |
| Electron exposure (e−/Å²) | 66 | | |
| Defocus range (μm) | 0.8–2.5 | | |
| Pixel size (Å) | 1.1 | | |
| Total movies acquired/used | 5187 | | |
| Symmetry imposed | C1 | | |
| Initial particle images (no.) | 3,835,569 | | |
| Final particle images (no.) | 456,948 | 64,449 | 64,449 |
| Map resolution (Å) | 3.21 | 3.4 | 3.4 |
| FSC threshold | 0.143 | 0.143 | 0.143 |
| Map resolution range (Å) | 2.8–3.8 | | |
| **Refinement** | | | |
| Initial model used (PDB code) | 5EJK | 7JN3 | 7KU7 |
| Model resolution (Å) | 3.1/3.2/4.0 | 3.0/3.4/4.1 | 3.0/3.4/3.9 |
| FSC threshold | 0/0.143/0.5 | 0/0.143/0.5 | 0/0.143/0.5 |
| Map sharpening B factor (Å²) | −30 | −30 | −30 |
| Map CC | 0.63 | 0.80 | 0.86 |
| Model composition | | | |
| Non-hydrogen atoms | 15716 | 15766 | 10630 |
| Protein residues | 1811 | 1807 | 1161 |
| DNA residues | 68 | 68 | 68 |
| Metal ions (Zn++, Mg++) | 2, 4 | 2, 4 | 2, 4 |
| Ligand (MK-2048) | 2 | 2 | 2 |
| B factors (Å²) | | | |
| Protein | 86.59 | 457.12 | 184.14 |
| DNA | 108.33 | 243.07 | 226.14 |
| R.m.s. deviations | | | |
| Bond lengths (Å) | 0.004 | 0.005 | 0.006 |
| Bond angles (°) | 0.730 | 0.980 | 1.127 |
| Validation | | | |
| MolProbity score | 2.22 | 2.20 | 2.30 |
| Clashscore | 16.42 | 17.61 | 21.02 |
| Poor rotamers (%) | 0.07 | 0 | 0.10 |
| Ramachandran plot | | | |
| Favored (%) | 91.66 | 92.88 | 92.14 |
| Allowed (%) | 8.34 | 7.12 | 7.86 |
| Disallowed (%) | 0.0 | 0 | 0 |

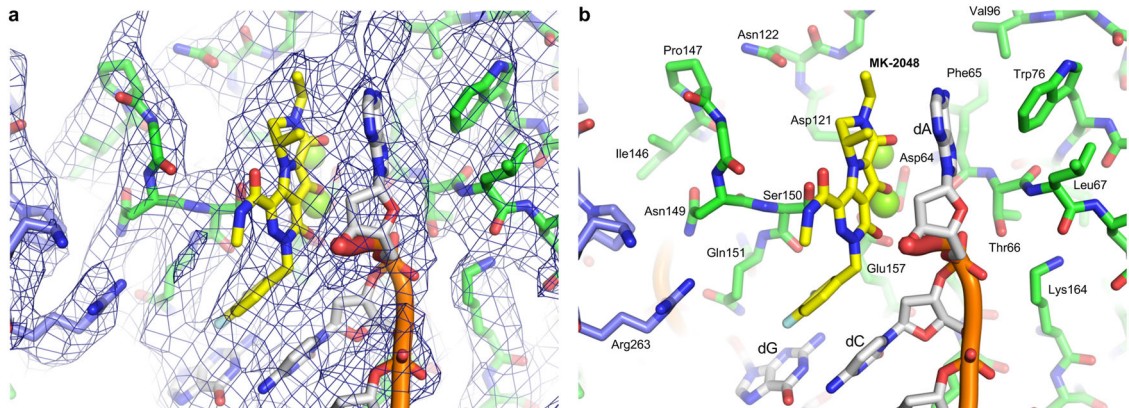

**Fig. 3 MK-2048 in the RSV IN active site. a** A zoomed view around the active site of the catalytic IN molecule occupied by MK-2048. The protein residues and viral DNA nucleotides are shown as sticks, with the viral DNA backbone path shown as orange tubes. The two catalytically important magnesium ions coordinated by active site residues Asp64, Asp121, and Glu157, are represented by light green spheres, shown at 50% of the van der Waals radius. The cryo-EM density contoured at 8.0 sigma level in blue mesh is overlaid on the model. **b** Same view as in (**a**), with the protein residues and DNA bases labeled.

of PFV, Tyr212, is instead involved in a hydrophobic interaction with EVG. Our results provide a structural understanding of EVG resistance to RSV replication, concerted integration activity, and CSC intasome assembly.

**Flexibility of distal IN dimers in the RSV CSC in comparison to the CIC.** The NTDs and CCDs of the distal IN dimers have weaker and less well-defined densities compared to the rest of the CSC. To evaluate this flexibility, a three-dimensional variability

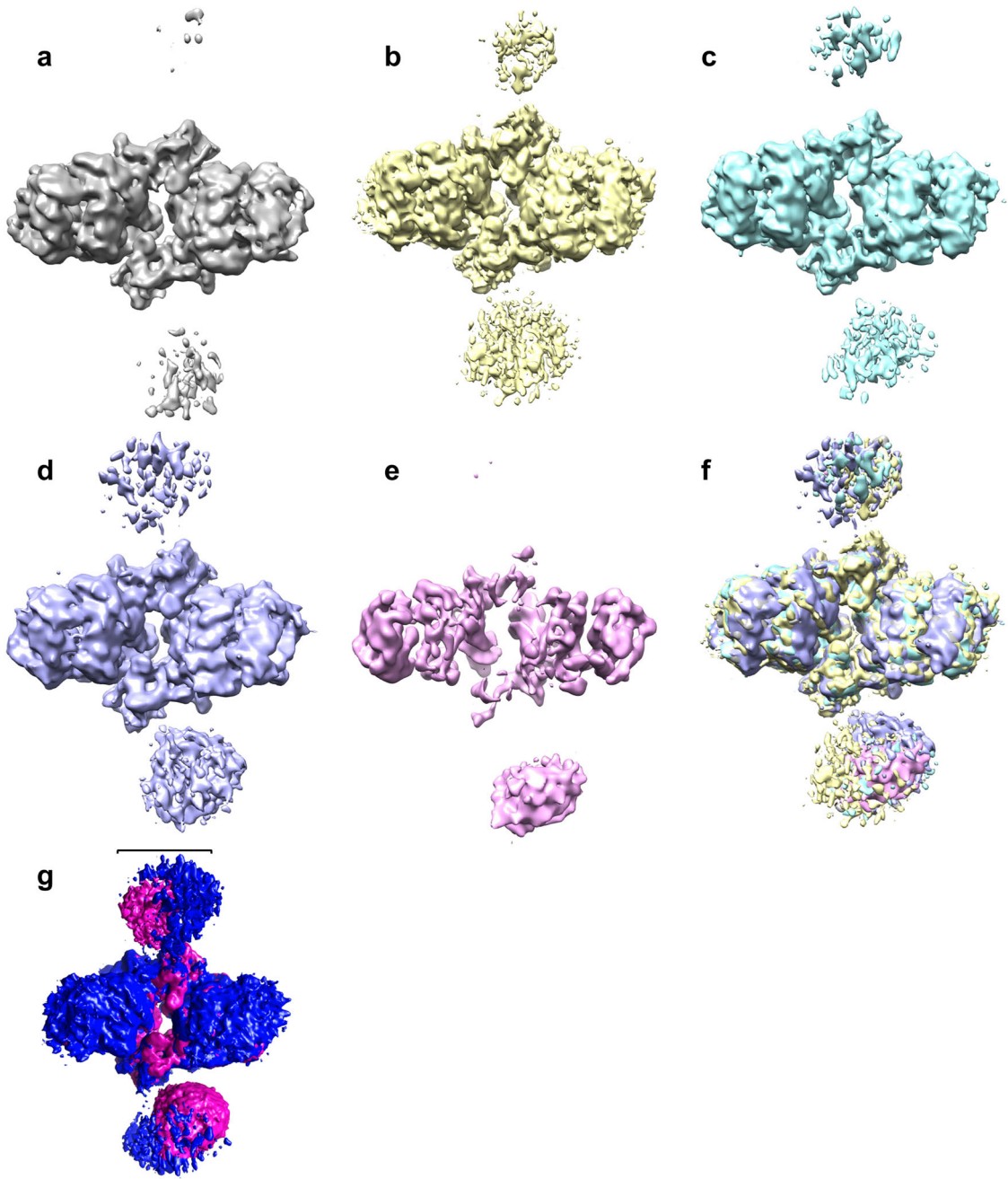

**Fig. 4 Three-dimensional variability analysis of the RSV CSC. a–e** 3D reconstructions of five clusters at a constant threshold level. Overlay of these five clusters is shown in (**f**). **g** Frames overlay of two reconstructed RSV CSC maps (shown in magenta and blue) generated by 3D variability analysis. The movement in flexible distal subunits is shown in brackets (See Supplemental Movie 1).

analysis (3DVA) was performed in cryoSPARC[23,24] at a low-pass filter resolution of 7 Å with three variability components, which demonstrated continuous sampling of conformations in the CSC. The NTD-CCD regions of the distal dimers showed maximum flexibility, which likely contributed to their comparatively poor local resolutions compared to the CIC region (Supplementary movie 1). Variable conformations in distal dimers showed ~20–30 Å movement (Fig. 4g). The outer subunits in the proximal dimer have a relaxed conformation demonstrating an outward concerted movement of at least 5 Å when the distal NTD-CCD is further away from the CIC. When the distal NTD-CCD is closer to the proximal dimers, it makes the proximal dimers more compact. The particles were grouped into five

clusters (A–E) in their reaction coordinate space. The particles from each of these clusters were used to perform 3D reconstruction (Fig. 4a–e). Two of these clusters (Fig. 4a, e) had minimal NTD-CCD density for one of the distal dimers; other clusters also demonstrated variability mainly in the distal NTD-CCD region. We further refined these clusters by re-extracting the particles at maximum resolution and performed 3D reconstruction followed by homogeneous refinement. The refined structure for cluster E, with 64,449 particles, displayed the highest resolution (3.4 Å) (Supplementary Fig. 9c) (Table 1). Comparison of the model generated with this refined cluster and the CSC intasome demonstrated the conformational heterogeneity in the distal subunits (Supplementary Fig. 9e). These observations are

consistent with a more relaxed CIC structure in the RSV CSC as compared to that in the STC, and highlights the overall dynamic potential of the RSV intasome. We accordingly speculate that distal IN dimer NTD-CCD flexibility may facilitate target DNA capturing. Conversely, the binding of target DNA through its interactions with distal CCDs may stabilize the inherent flexibility of the distal dimers, leading to a more rigid STC structure.

**Functional CTD and "tail" region boundaries for CSC assembly and RSV infectivity.** We next wanted to understand the interactions between the CTD and the "tail" region of IN to promote the assembly of intasomes and to foster virus infectivity. CTDs play critical roles in stabilizing functional retrovirus intasome CICs[1,3,4,6,8]. The RSV IN "tail" region (defined as residues from Ile269 to Ala286; Supplementary Fig. 3) accelerates the conversion of the precursor tetrameric CSC to the mature octameric form in a time and temperature-dependent manner[15,16]. RSV IN 1–269 catalyzes 3′ OH processing of blunt-ended viral DNA at wt IN levels and concerted integration into supercoiled target DNA at ~70% level of wt IN[14]. IN 1–269 tetrameric CSCs could be isolated only in the presence of an INSTI[14] while IN 1–278 octameric CSCs could be isolated in the presence (Supplementary Fig. 2) or absence of INSTIs (Supplementary Fig. 10).

We first investigated the role of the IN "tail" region in virus infection. We mutagenized the corresponding region of pRIAS-Luc, a single-round reporter virus that expresses IN composed of 323 residues, which is processed to IN 1–286 by viral protease in the virus[25–27], and firefly luciferase as a readout for RSV infectivity[20]. A series of truncation constructs were initially built that progressively shortened the IN tail (Fig. 5a, terminal residues marked in red). The quantity of virus produced from transfected cells was determined using an exogenous reverse transcriptase (RT) assay with virion lysates, and cells were infected with normalized levels of wt IN 1–286 and/or IN 1–323 RT units alongside IN mutant viruses. This analysis revealed that the tail region of IN was dispensable for RSV infection (Fig. 5b).

Prior analyses of the HIV-1 IN tail region, which is similar in length but different in sequence from RSV, revealed somewhat contrasting results (Fig. 5a)[28,29]. Whereas results of one study indicated residues downstream from Asp270 (analogous to RSV IN residue Thr270) in HIV-1 were critical to support the infection of a single-round reporter virus[28], the results of a separate study revealed that a virus terminating at Arg269 (analogous to RSV IN residue Ile269) supported efficient HIV-1 replication[29]. In both HIV-1 studies, truncations that extended into the conserved β10 secondary structural element of the CTD ablated single-round infection and spreading HIV-1 replication (Fig. 5a). We accordingly next tested infectivities of RSV IN derivatives 1–266 and 1–264 that partially and fully deleted β10 sequences, respectively. Not unexpectedly, both RSV IN mutants were noninfectious (Fig. 5c) demonstrating that an intact CTD is necessary for infectivity.

Our C-terminal truncation analysis revealed all-or-none viral infection phenotypes. Unfortunately, recombinant proteins corresponding to IN deletion mutants 1–266 and 1–264 were insoluble upon expression in bacteria and could not be characterized biochemically. To further investigate the functionality of the recombinant IN CTD/tail mutant proteins and the activity of corresponding mutant viruses, we turned to missense mutagenesis. Due to the importance of β10 for RSV/HIV-1 infection, we targeted both conserved and variable residues that comprised the η3 helix that lies immediately upstream from β10 (Supplementary Fig. 3). IN mutant viruses R263A and R263K, which altered conserved RSV IN residue Arg263, supported ~20% and 50% of the level of wt 1–323 infection, respectively (Fig. 5c).

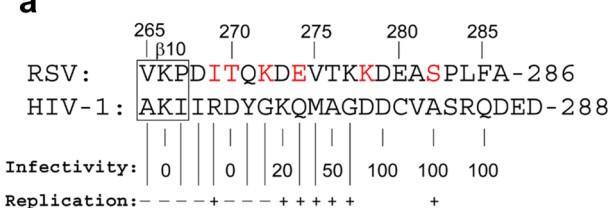

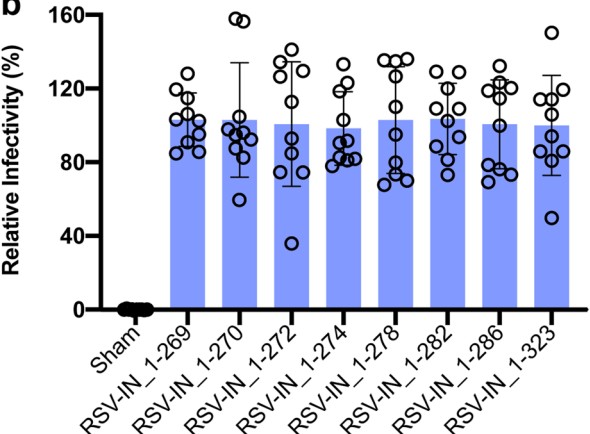

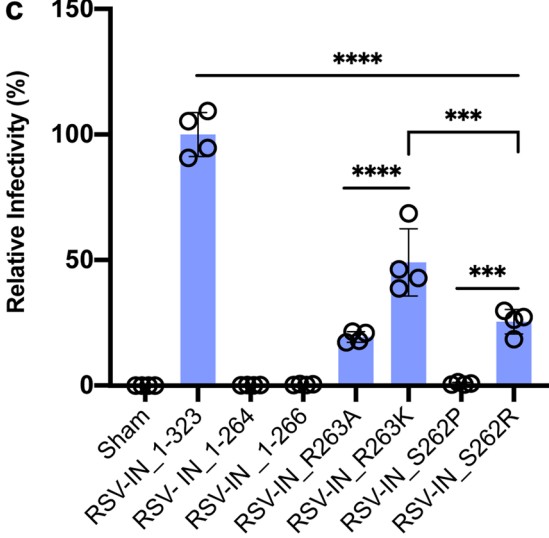

**Fig. 5 Single-round RSV infectivity assays. a** Sequence alignment of the RSV/HIV-1 IN tail region. The RSV IN terminal residue for each truncation is marked in red. The bottom HIV-1 IN residues are marked by percent of infectivity[28] and replication[29] of each truncation compared to wt virus. **b** Equivalent quantities of RSV virions produced by the indicated pRIAS-Luc constructs were used to infect DF-1 cells. The length of wt IN is 286 residues. C-terminal truncated IN mutants are indicated by the last residue in the construct. IN 1–323 has the natural 37 amino acid extension that is cleaved off in virions. Results (average ± standard deviation) compile data from five independent infection assays, each conducted with duplicate samples; data points show results of all technical replicates. **c** Infectivities of C-terminal IN truncations (1–264 and 1–266) and four missense IN mutants (R263A; R263K; S262P; S262R) versus pRIAS-Luc (IN 1–323) are shown (average ± standard deviation for two independent experiments, each conducted with technical duplicates). Sham samples were media harvested from DF-1 cells cultured under identical conditions except plasmids were omitted from transfection mixes. Statistical significance between samples was assessed using one-way ANOVA in GraphPad Prism version 8. ***$p < 0.0001$; ****$p < 0.00001$.

Both purified RSV IN mutants were unable to form the octameric CSC while IN R263K was only capable of producing the tetrameric CSC[16] suggesting that Arg263 in the distal IN dimer may play an important role in octameric intasome assembly (Supplementary Fig. 6b). While η3 helix IN mutant virus S262P that targeted variable RSV IN residue Ser262 was noninfectious, S262R displayed ~25% of wt infectivity (Fig. 5b), similar to what was previously observed under conditions of spreading RSV replication[30]. Recombinant IN S262P was defective for CSC assembly while S262R formed octameric CSC intasomes, albeit at lower efficiency than wt IN (Supplementary Fig. 11).

IN and RT β subunit (precursor to IN) content of wt and mutant virions were probed by immunoblotting viral lysates alongside purified IN and RT β subunits proteins as controls[27,31] (Supplementary Fig. 12). IN was undetectable in noninfectious viruses IN 1–264, 1–266, and S262P; S262P also lacked detectable RT β protein. Partially infectious mutants IN R263A, R263K, and S262R expectedly harbored IN protein alongside variable amounts of RT β. These data suggest that the instability of the β subunit and IN mutant proteins 1–264, 1–266, and S262P likely contributed to the noninfectious phenotypes of these viruses (Fig. 5c).

## Discussion

Cryo-EM structural studies of the RSV octameric CSC stabilized by INSTI MK-2048 have provided important insight into previously unappreciated dynamic properties within the CIC as well as complexities associated with the assembly of retroviral intasomes. It is notable that the RSV octameric CSC produced with viral DNA and the STC produced with viral/target DNA substrate[5] show significant differences in the positioning and flexibility of the non-catalytic distal IN dimers (Fig. 2a, b). The distal IN subunits display a high degree of flexibility in the CSC (Fig. 4) (Supplementary Fig. 9, Supplementary movie 1). Further analysis of the octameric CSC cryo-EM data defines the high flexibility of the tethered NTD-CCD domains of the distal IN dimers that may help to capture the target DNA for integration. The sampled conformations of the distal IN dimers in the CSC are more open and distinct from those observed in the STC (Fig. 2c), suggesting that the distal IN CCDs do not dictate the trajectory of the sharply bent target DNA in the STC. Rather, these domains are likely to adopt the conformation observed in the STC structure by following the path of the target DNA (i.e., target DNA bending is induced by the catalytic CCD of the proximal IN dimers). It is understood that the RSV STC crystal structure reveals a static view for a specific conformation while the cryo-EM structures presented here demonstrate an ensemble of conformations reflecting the overall flexibility in the CSC before target DNA binding. Further structural work, for example via solving the RSV STC structure by cryo-EM, will be needed to ascertain how the CSC converts to the STC upon target DNA binding and how target DNA stabilizes the flexible NTD-CCDs in the distal subunits. In earlier structural studies of PFV[1,19] and MVV[10] intasomes, which are tetrameric and hexadecameric IN assemblies, respectively, showed essentially no changes before and after target DNA binding. These differences may reflect limited flexibilities of PFV and MVV intasomes as they morph between transitional states, which could potentially affect integration target site preferences between the different retrovirus genera and their association with cellular cofactors including LEDGF/p75 for lentiviruses[17]. In summary, these studies highlight complexities associated with diverse assembly mechanisms of retroviral intasomes, even though the catalytic integration mechanisms of all retroviruses are very similar.

The ability of RSV IN to assemble the catalytically active octameric CSC from its tetrameric CSC precursor affords opportunities to investigate this distinct intasome assembly pathway[14–16]. The CTD of the distal IN dimers in the CSC plays a critical structural role together with the proximal IN dimers to stabilize the CSC (Figs. 1 and 2). Similarly, the distal IN (flanking) dimers of the MMTV octameric CSC are also flexible and these CTDs likewise play a critical role to form and stabilize the CIC[6]. As expected, the two catalytic RSV IN molecules swap their NTDs across the synaptic interface (Fig. 1a, d)[18]. The precursor to the RSV tetrameric CSC is unknown. However, fluorophore resonance energy transfer analysis of both the tetrameric and octameric CSC intasomes demonstrated that both contain two viral DNA molecules with their 5′-ends in close proximity[15]. Trans-communication can occur between wt IN dimers bound to two different 3′ OH recessed viral DNA ends, i.e., an IN dimer bound to GU3 can couple with another IN dimer bound to a defective U3 end that by itself is incapable of assembling the CSC[16]. We speculate that the assembly process occurs in a stepwise fashion, possibly by the formation of the tetrameric intasome from two independent proximal IN dimers bound to viral DNA ends, followed by binding of the flexible distal dimers donating the critical CTDs into the CIC of the mature octameric CSC.

Truncations of the C-terminal tail region (17 residues) of RSV IN (Supplementary Fig. 3) determined that these residues were unnecessary for infection using single-round infectivity assays (Fig. 5b), 3′ OH processing of blunt-ended viral DNA and concerted integration activity in vitro[16]. To expand the correlation of CTD/tail mutant activities in vitro to virus infection beyond the all-or-none phenotypes of C-terminal truncations mutants, we turned to missense mutagenesis of the η3 helix that lies immediately upstream from β10 (Supplementary Fig. 3). The results of these analyses demonstrated that virus infectivity (Fig. 5b)[30] and assembly of either the octameric and/or tetrameric intasomes were affected in parallel (Supplementary Fig. 11)[16]. The C-terminal tail truncated RSV IN 1–269 possessing the CTD accumulates tetrameric intasomes, suggesting that the C-terminal tail region plays a role in octameric intasome assembly in vitro[14,15]. Plausibly, the requirement for the C-terminal tail is compensated in the cellular context by the flexible distal INs that contribute the critical stabilizing CTD to a tetrameric complex, thus forming the mature octameric intasome. Because IN 1–269 efficiently assembles the octameric STC, host/target DNA is a potentially compensatory element for the tail region in infected cells[5,15]. Investigations into the assembly, structure, and functional properties of the RSV tetrameric intasome warrant further efforts.

## Methods

**RSV IN expression and purification**. RSV IN constructs were expressed in *Escherichia coli* BL21 (DE3)pLysS and purified to near-homogeneity as previously described[14–16]. The wt Prague A IN subunit is 286 aa in length, designated 1–286. Dimeric C-terminal truncated IN (1–278 residues) was purified and concentrated to 20–30 mg/ml using Amicon Ultra-15 centrifugal filters. The protein concentrations were expressed as monomeric subunits.

**Concerted integration assays**. The assay conditions for concerted integration of viral DNA ends into supercoiled target DNA were previously described[15,16]. Double-stranded 3′ OH recessed oligonucleotide (ODN) containing RSV gain-of-function(G) U3 long terminal repeat (LTR) sequences were 18 nucleotides in length. The sequence of the GU3 18R substrate on the 3′ OH recessed catalytic strand is 5′-ATTGCATAAGAC**A**ACA-3′ and the complementary non-catalytic strand is 5′-AATGTT**G**TCTTATGCAAT-3′. The bold nucleotide A on the catalytic strand of GU3 is T in the wt U3 substrate. The octameric CSC was assembled in the absence of MK-2048 at 14 °C to allow measurement of concerted integration activity upon purification of the CSC by size exclusion chromatography (SEC) (Supplementary Fig. 10). The GU3 substrate was used for assembly of the octameric RSV CSC for EM negative staining and cryo-EM imaging. Assays for measuring concerted integration activity in the presence of varying amounts of

INSTIs were previously described[14]. MK-2048 was generously provided by Merck & Co.

**Assembly and purification of the RSV octameric CSC**. The RSV octameric CSC was assembled with IN 1–278 and GU3 18R in the presence of MK-2048[14–16]. The assembly buffer was 20 mM HEPES, pH 7.5, 150 mM NaCl, 1 M non-detergent sulfobetaines (NDSB)−201, 10% dimethyl sulfoxide (DMSO), 10% glycerol, 50 mM MgSO4, and 1 mM tris(2-carboxyethyl phosphine) (TCEP). IN (as monomers), 3′ OH recessed DNA ODN and INSTI concentrations were 45, 15, and 125 μM, respectively, unless otherwise indicated. After the addition of MK-2048 and DNA, IN was added to the assembly mixture. The samples were generally incubated at 18 °C for 18 h. The octameric RSV CSC was purified by SEC using Superdex 200 Increase (10/300)(GE Healthcare Life Sciences)[16] (Supplementary Fig. 2). The SEC running buffer was 20 mM HEPES pH 7.5, 100 mM NaCl, 50 mM MgSO4, and 1 mM TCEP. INSTI was omitted from the running buffer. Chromatography was at 4 °C and UV absorption monitored at 280 nm. Generally, 100–450 μl of the sample was injected into the column depending on whether the samples were for concerted integration activity, EM negative staining, or cryo-EM imaging. The samples were immediately used for these procedures after SEC purification. Re-chromatography of pooled CSC fractions on Superdex 200 established that the octameric RSV CSC was stable on ice for at least 4 h. The concentration of the CSC was determined by quantitative SDS-PAGE with Coomassie Fluor Orange (Invitrogen) protein gel stain.

**EM negative stain grid preparation, data collection, and processing**. The RSV octameric CSC, freshly purified by SEC and generally between 0.015 and 0.05 μM, was used for negative staining. The grid (carbon-coated 200 mesh copper grids) was placed on a 10 μl drop of purified CSC and incubated for 1 min at room temperature. Prior to applying the sample, the grid was glow-discharged for 1 min at 25 mA using a GloQube glow-discharger. Post-incubation, the grid was washed serially in water drops and stained with 0.75% uranyl formate for 3 min. The grid was blotted using filter paper and air dried. The grids were imaged on a JEOL 1400 TEM equipped with an AMT CCD camera operating at 120 kV at 80,000 nominal magnification resulting in a magnified pixel size of 2.30 Å. The particles were picked automatically using Blob picker and 17,961 particles were used for reference-free 2D classification in 50 class-averages. The 9 class-averages representing 9976 particles were selected for Ab initio 3D reconstruction. Each of the two 3D classes was refined and sharpened 3D maps constructed (Supplementary Fig. 4).

**Cryo-EM sample preparation and imaging**. Cryo-EM samples were derived from the peak SEC fractions without dilution (~0.5–0.7 μM). The samples were prepared on quantifoil holey carbon grids (R2/2 300 mesh copper), which were plasma cleaned for 1 min using a Gatan Solarus 950 (Gatan, Pleasanton, CA), and plunge frozen using a Vitrobot Mark IV (ThermoFisher Scientific, Brno, CZ). The Vitrobot sample chamber was set to 4 ºC and 100% humidity. Three microliters of purified RSV CSC were applied to the plasma cleaned quantifoil grids and allowed to incubate for 20 s. Samples were then blotted for 2 s at a blot force of −1 and plunge frozen into liquid ethane.

Vitrified grids were imaged using a Cs-corrected Thermo Fisher Titan Krios G3 electron microscope (ThermoFisher Scientific, Brno, CZ) operating at an accelerating voltage of 300 kV equipped with a Gatan K2-Summit detector (Gatan, Pleasanton, CA) and a BioQuantum 968 GIF-quantum energy filter (Gatan, Pleasanton, CA) operating with a slit width of 20 eV. Data acquisition was automated using EPU software (ThermoFisher Scientific, Brno, CZ) at a magnification of ×105,000 in super-resolution mode which corresponds to a pixel size of 0.55 Å (physical pixel size of 1.1 Å). Micrograph movies were recorded for 8 s with 40 frames. This results in a frame rate of 0.2 s per frame with a dose rate of 1.65 electrons per Å² per frame (total dose of 66 electrons per Å²). The defocus was varied between −1 and −2.5 μm. In total, 5187 movies were recorded in two separate sessions of two days each.

**Cryo-EM image analysis**. The movies were corrected for beam-induced movement using MotionCorr2[32]. Further data analysis was done in cryoSPARC 2.9[24]. The contrast transfer function (CTF) was determined using Patch-based CTF estimation. Initially, particles were picked from 1000 images and reference-free 2D classification performed. The selected 2D classes were used as a template to pick particles from all images and 2D classification was performed in 100 classes. After removing the junk particles, 2D class-average containing ~1.8 million particles were used for ab initio 3D reconstruction in five classes. The 3D reconstructions were refined by homogeneous refinement without imposing symmetry. Using C2 symmetry during the refinement did not significantly improve the map resolution possibly due to the flexible nature of the complex. One of the refined 3D classes representing the best resolution and highest number of particles (456,948) was sharpened and used for model building (Supplementary Fig. 5e). Gold standard FSC value of 0.143 was used to determine the resolution of 3D reconstructions. Heat map of the angular distribution of refined particles used in reconstruction is shown in (Supplementary Fig. 5f). Directional FSC curves and map anisotropy were calculated in cryoSPARC and remote 3DFSC server[33]. The

local resolution in the CIC region was determined in cryoSPARC and displayed in Chimera (Supplementary Fig. 5i). The local resolution map demonstrated ~2.8–4.0 Å resolution in the core region. Part of cryo-EM imaging data was also independently processed in RELION 3.0[34] and yielded a similar 3D reconstruction. The 3D variability analysis (Fig. 4) (Supplementary movie 1) was performed at a low-pass filter resolution of 7 Å with three variability components to determine the conformational heterogeneity and flexibility of the IN dimers in the octameric RSV CSC. The particles were grouped into five clusters in their reaction coordinate space. The particles in each cluster were reextracted at maximum resolution and used for 3D reconstruction. Each cluster was refined by homogeneous refinement. One of the cluster (E) map yielding the highest resolution was used for model building (Supplementary Fig. 9).

**Preparation of the atomic model, refinement, and validation**. The starting model of RSV CSC was obtained by removing the target DNA from our previously reported RSV STC crystal structure (PDB ID: 5ejk). The preliminary model was docked into the EM map as a rigid body and manually modified/rebuilt using COOT[35]. The preliminary model thus obtained was refined using PHENIX real_space_refine[36,37] against the cryo-EM density and a standard set of geometry/stereochemistry restraints. The stereochemistry restraints for MK-2048 were generated using eLBOW[38] in PHENIX Suite[39].

**Docking of INSTIs**. The molecular docking of EVG in the octameric RSV CSC model was performed using AutoDock VINA[40] (Supplementary Fig. 7). Docking was done using the inner subunit of proximal dimer bound to MK-2048 and viral DNA along with Mg++. The S150P substitution was generated in silico using Chimera[41] and it did not result in clashes with neighboring residues (Supplementary Fig. 8). The EVG structure was built in Chimera. The polar hydrogen atoms were added to the structure. The best pose sorted on the basis of VINA score and INSTI interactions in the active sites was selected. Docked structures were displayed in Discovery Studio Visualizer (BIOVIA).

**Plasmids for RSV infectivity assays**. A two plasmid system for transfection was used to produce infectious pseudotyped RSV virions with pRIAS-Luc[20] and the G envelope glycoprotein of the vesicular stomatitis virus (pHCMV-VSV-G)[42]. Plasmids were constructed that contained C-terminal truncations of RSV IN or single-point mutations in the C-terminal region. The desired mutations in the IN region of the *pol* gene were synthesized and inserted back into the original pRIAS-Luc plasmid (Genscript). The original pRIAS-Luc construct contained RSV IN (1–323 aa). IN 1–323 has the natural 37 amino acid C-terminal extension that is cleaved off in virions producing wt IN (286 aa in length)[25,43]. This extension is not required for virus replication[26]. We investigated the effects of IN modifications within pRIAS-Luc vectors using viral infectivity assays (Fig. 5). Each C-terminal IN truncation is identified by the last amino acid in the IN sequence prior to the stop codon. The different pRIAS-Luc constructs are pRIAS-RSV-Luc (IN 1–323); (IN 1–286); (IN 1–282); (IN 1–278); (IN 1–274); (IN 1–272); (IN 1–270); (IN 1–269); (IN 1–266) and (IN 1–264). Single-point mutations were introduced into pRIAS-Luc (IN 1–323). They are IN S262P; S262R; R263A and R263K. The sequences of all plasmids were verified by DNA sequence analysis.

**Cells and viruses**. DF-1 cells[44] which were used both to make viruses by transfection and for infection studies, were propagated at 37 °C in a humidified incubator with 5% CO2 in Dulbecco's-modified Eagle medium (DMEM) supplemented to contain 10% (v/v) fetal bovine serum, penicillin (100 IU/mL), and streptomycin (100 μg/mL)(DMEM+). Cells were co-transfected with pRIAS-Luc and pHCMV-VSV-G plasmids at mass ratios from 3:1 to 9:1 using Polyjet In Vitro Transfection Reagent (SignaGen). Virus-containing cell supernatant harvested twice over a 3-day period (at 48 and 72 h) was filtered through 0.45 μm filters and concentrated by ultracentrifugation at 26,000 × g for 2 h. Samples resuspended in DMEM were stored in −80 °C. Virus yield was assessed by Taqman-based product-enhanced RT (Taq-PERT), which measures virion-associated RT activity, essentially as previously described[9]; standard curves, which generated mU/mL values, were constructed using recombinant HIV-1 (Millipore Sigma) or avian myeloblastosis virus (AMV) dimeric αβ RT protein (Amsbio LLC). Cells were infected in duplicate using matched wt and IN mutant inocula (500–2000 mU 100,000 cells)(Fig. 5). After 5 h, cells washed with phosphate-buffered saline (PBS) were replenished with fresh DMEM+, followed by incubation for 43 h for a total 48 h infection time course. Cells lysed using Passive Lysis Buffer (Promega Corp.) were processed in duplicate for luciferase assays. Resultant relative light units (RLUs) were normalized to total protein concentration in cell extracts as determined by Pierce bicinchoninic acid protein assay kit (Thermo Fisher Scientific).

**Western blot analysis**. Rabbit polyclonal antibodies directed against a 14 amino acid peptide derived from sequences near the N-terminus of RSV IN or the purified AMV full-length IN eluted from an SDS-gel were previously described[45]. These antisera detected the β subunit of RT and IN in virions by immunoprecipitation and Western blotting[30]. All of the recombinant C-terminal IN truncations and single-point IN mutants and the β subunit of purified αβ RT were detected by Western Blotting with the above IN antisera. Virus pellets re-suspended in 1X lysis

buffer (0.125% Triton X-100, 25 mM KCl, 50 mM Tris-HCl pH 7.4, 20% glycerol) were assessed for IU/mL values by Taq-PERT as described above. Viruses matched for RT mU content and recombinant proteins were fractionated through 4–12% Bis-Tris polyacrylamide gels in MOPS SDS running buffer (Thermo Fisher Scientific) at 120 V (Supplementary Fig. 12). Proteins were transferred to polyvinylidene difluoride membranes using Bis-Tris transfer buffer (Thermo Fisher Scientific) containing 8% methanol at 90 V for 1 h. Membranes were subsequently blocked using 5% milk in PBS-0.05%, Tween 20 (PBST) for 1 h at room temperature.

Primary rabbit antibodies were incubated with membranes at 1:5,000 dilution overnight in 2.5% milk–PBST at 4 °C. Membranes washed three times with PBST were probed with secondary horse radish peroxidase-conjugated goat anti-rabbit antibodies (Dako Products) at 1:40,000 dilution in 2.5% milk–PBST for 1 h at room temperature. Membranes washed three times with PBST and once with PBS were developed using a mix of supersignal west pico plus chemiluminescent substrate and supersignal west femto maximum sensitivity substrate (Thermo Fisher Scientific). Immunoblots were imaged using a ChemiDoc MG Imager (Bio-Rad) with Image Lab v.5.2.1.164. Membranes stripped using Restore Western Blot Stripping Buffer (Thermo Fisher Scientific) were washed three times with PBS and blocked with 5% milk–PBST for 1 h at room temperature. Stripped membranes were processed for immunoblotting as described above.

**Reporting summary**. Further information on research design is available in the Nature Research Reporting Summary linked to this article.

## Data availability

The cryo-EM maps were deposited with the Electron Microscopy Data Bank (accession code EMDB-22400 and 23035) and the refined model with the Protein Data Bank (7JN3, 7KU7, and 7KUI). All materials used in the manuscript are available upon request. The raw data used to prepare Fig. 5b, c are available as Supplementary Data 1.

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

## Acknowledgements

This work was supported by NIGMS R35-GM118047 (to H.A.), NIAID R21-AI127196 (to D.P.G.), Saint Louis University Research Growth Fund (to D.P.G.), NIAID R01-AI070042 (to A.N.E.), the Children's Discovery Institute of Washington University, and St. Louis Children's Hospital, CDI-CORE-2015-505 and CDI-CORE-2019-813 (to J.A.J.F.) and the Foundation for Barnes-Jewish Hospital, 3770 (to J.A.J.F.).

## Author contributions

S.B. purified the recombinant proteins and the CSC for cryo-EM. K.K.P. and S.B. carried out the biochemical analysis and concerted integration assays. M.J.R. performed the negative staining, prepared the cryo-EM grid, and collected imaging data on Krios. M.J.R. and J.A.J.F. carried out initial cryo-EM data analysis using RELION. K.K.P. analyzed the cryo-EM data for single-particle reconstruction and carried out other computational analysis. K.S. and H.A. performed model building and refinement. K.K.P. and A.N.E. designed the constructs for infectivity assays. A.V.O. and A.N.E. performed the virus infectivity assays and Western Blot analysis. K.K.P., H.A., A.N.E., and D.P.G. wrote the manuscript with contributions from all authors. D.P.G. managed the project.

## Competing interests

A.N.E. over the past 12 months has received fees from ViiV Healthcare Co. No other authors declare competing interests.
