## [Peer Review File · Communications Biology]

Reviewers' comments:

Reviewer #1 (Remarks to the Author):

In this manuscript, Pandey et al determine the cryo-EM structure of the RSV cleaved synaptic complex (CSC) intasome bound to the experimental inhibitor MK-2048. They compare the structure of the CSC intasome to an X-ray structure of the strand transfer complex (STC) intasome determined previously, and through this comparison, they speculate on the molecular mechanisms underlying viral intasome assembly and function, an important area of investigation in the field. Their major claim is that the distal integrase (IN) dimers are flexible in the CSC but rigid and bound to target DNA in the STC. They conclude from these differences that the distal IN dimers rigidify upon target DNA capture and promote the formation of the target DNA-bound complex, i.e. that there is an ordered conformational transition between the key intermediates along the RSV integration pathway. While this might be the case, and is indeed one logical way of thinking about an assembly mechanism, the experimental data currently does not unambiguously support this claim. Most importantly, there is a fundamental issue with comparing the RSV CSC obtained by cryo-EM with the RSV STC obtained by X-ray crystallography. In the former, the CSC is in solution, and is therefore free to adopt a wide variety of conformational arrangements (some of which are observed in the current work). In the latter, the STC is locked into a single conformation through crystallographic packing. In most cases, I would not have an issue comparing cryo-EM to X-ray structures, for example when these authors discuss the configuration of the drug MK-2048. However, the real focus of the current work is specifically on dynamics and structural transitions, and therefore the comparison of the two structures is akin to comparing apples to oranges. Clearly, the authors understand this point (page 10, line 235), but chose not to address it prior to submission of the work. This, to me, is a sufficiently critical limitation, and therefore warrants obtaining either a cryo-EM structure of the STC or an X-ray structure of the CSC prior to making the current conclusions. For this reason, I believe that major revisions are required for this manuscript in its current form, and the authors would need to obtain such experimental data.

The idea that RSV intasomes transition from a tetramer to an octamer assembly has been argued previously in several papers, and the authors hypothesize that a tetrameric intasome with two independent IN dimers bound to two vDNA ends is a precursor to the bona fide octameric intasome. While prior biochemical data partially support this view, there is no structural evidence for this transition, and therefore this assembly mechanism remains speculative. An alternative explanation is that the tetrameric INSTI-trapped intasome actually represents an individual "half-intasome" (with one vDNA). This might still be an on-pathway intermediate, but would not require dramatic structural rearrangements for transitioning into and forming the complete octameric RSV intasome. Please remove specific language that suggests that the authors have a thorough understanding of what the tetrameric "intasomes" represent.

Additional minor comments are below:

Figure 1. Why is the model shown before the map?

Figure 3. This figure needs to be improved. These are raw snapshots from Chimera, and even the colors haven't been changed from their defaults. Some way of synthesizing this information would go a long way.

Supplementary Figure 1. "present" should be "presence"

Supplementary Figure 2. Please show a gel as well.

Supplementary Figure 4. The half-map and map-model FSC curves should be shown on the same scale for easier comparison. It looks like the model only correlates to the map up to a resolution of $\sim 1/0.26$ (~ 3.8 Å), whereas the map is resolved to ~ 3.2 Å. Ideally, these should be very similar. This discrepancy should be addressed. Either (1) the model would need to be improved, or (2) the FSC curves need to be obtained and compared for the most homogeneous regions of the map (presumably CIC). Ideally, one would also show 3D FSCs.

Supplementary Figure 5. How was local resolution calculated? It looks broken up and not very smooth, as one would expect.

Supplementary Figure 10. How does the activity of the tetrameric CSC or IN alone compare with that of the octameric CSC in concerted integration?

Supplementary Figure 13. This is negative stain or cryo? If negative stain (as in the legend), why is the contrast reversed? Why is this figure last? If negative stain, presumably this should be shown before the cryo-EM data.

Table S1.

- Why was no symmetry imposed?
- Clashscore is quite high. Please address

Reviewer #2 (Remarks to the Author):

This manuscript reports a cryo-EM structure of the RSV intasome prior to its engagement with target DNA. As such, the new data nicely complement and extend prior work by the same group that described the strand transfer complex (i.e. intasome plus target DNA, published in 2016). The current structure was determined in the presence of a strand transfer inhibitor MK2048. The structure plus mutational data seem to explain why some of these small molecules inhibit RSV replication (RAL, MK2048, etc), while some (EVG) do not. This part of the story is interesting, straightforward and arguably important.

The authors then decide to delve into the mystery of the C-terminal tail of RSV integrase. They have published at least two papers on this tail, and frankly the current work does not add any clarity. They do show that the tail is not essential for RSV replication, and as far as I understand, they do not see it in their reconstruction. I feel that too much emphasis is made on that tail, which distracts from the key findings. Annoyingly, it is difficult to understand this part of the manuscript without downloading the previous papers about the tail. In addition to adding clarity, it would help to include a panel (within a main figure) showing the sequence of the tail, indicating which tail residues are visible in the current (and previous?) structure.

Further points:

By the look of it, RSV intasome appears very similar to the cryo-EM reconstruction of the MMTV intasome (EMD-6441). It would be good to compare or at least mention.

Line 355: "homologous refinement"

The particle number in the final set is very large. Fig S4 does not show results of a 3D classification;

was it even done (beyond ab-initio)? Relion is much more robust in 3D classification than heterogenous refinement in Cryosparc, for example; and ab-initio is not very powerful as a 3D classification tool. My impression is that cryo-EM data processing was done without due care.

I would expect non-uniform refinement in Cryosparc to work better in this case, where considerable disorder (distal dimers) is present (this might not be the case, but have the authors attempted it?).

According to Table 1, 2-fold symmetry was not used during reconstruction. Was there a particular benefit from not imposing C2?

The micrograph in Fig S13A does not look like negative stain!

Reviewer #3 (Remarks to the Author):

The manuscript Pandey et al. is revolving around the structure of the Rous sarcoma virus (RSV) octameric intasome intermediate called cleaved synaptic complex (CSC) and the role of the integrase C-terminal tail region residues in the infectivity and intasome assembly.

Compared with the already published RSV intasome strand transfer complex (STC) structure, the CSC shows an unexpected flexibility of its distal protomers. The functional consequence of this dynamic flexibility is however not clear and remains an interesting question to further address.

As the assembly procedure requires the stabilization with an integrase strand transfer inhibitor (INSTI), the author could also explain the molecular mechanisms involved in RSV Elvitegravir (EVG) resistance.

Finally, using a pseudotyped RSV reporter system, the author showed that the last 17 residues "tail region" were dispensable for RSV infection, data consistent with their earlier report using in vitro integration assay (Bera et al. 2018).

I don't have any major comments on the quality of the data themselves. This is a beautiful intasome structure revealing an unexpected feature of the distal dimers.

However, I have some questions and comments that might need to be addressed before publication.

1. L119-120. Is there a technical or biological explanation as to why one distal dimer shows minimal density?

2. L185. Considering the apparent important role of the tail region in the kinetics of intasome assembly I felt a bit disappointed by the sole infection experiment. One could not rule out that although the relative infectivity is similar 43h post transduction, the kinetics/timing of integration might be different between the truncated IN mutants. This can be addressed by synchronizing the infection and adding RAL or DTG to the cells at different time points (See similar experiment: Zurnic et al. PLoS Pathog 2016).

These truncation mutants might also show a different integration site selection profile. This can be discussed.

3. Paragraph starting at L187. To be honest, I don't see the rationale behind this mutational analysis (S262/R263) to understand the function of the tail region during infection. Can you explain?

4. L231-234. I think the comparison as such is fair but we could argue that i) PFV intasome is fully tetrameric and do not require/have distal protomers outside the conserved intasome core and ii) MVV intasome was assembled using LEDGF/p75 which is a priori crucial for complex formation and might

prevent further flexibility. For these reason I think it would be important to mention the differences between the models.

5. L263-267. The extreme tail region of gammaretroviral integrase CTD is involved in the interaction with the cofactors Brd2-4. The interaction stimulates MLV in vitro integration activity. Moreover, ALV integrase binds to the FACT complex via its CTD, interaction that also stimulate in vitro integration. What is the status of RSV IN with regards to FACT? Is FACT a cofactor for alpharetroviral integrases or only ALV? Depending on the answer, biochemical analyses with the mutants/truncations including FACT might reveal some interesting features. Also, infections were done in DF-1 cells, do you see a different infection phenotype using a heterologous mammalian system to change the cellular context?

Minor concerns:

6. L53. spumavirus is the old taxonomy, since 2019 the PFV genus is "Simiispumavirus"

7. L61. "RSV IN is homodimeric". Is it meant "in solution"?

8. L92. "determined at 3.8A" add "resolution"

9. Sup figure 8 and 9. Some of the spheres and details are hard to see. Increasing the surface transparency might help.

10. L224. Typo: confirmation instead of conformation.

11. I found a bit difficult to keep track with the IN truncations and the point mutants in regards to their propensity to forms tetrameric vs octameric CSC vs STC, their in vitro activity and infectivity. Is it possible to create a table (as a figure or supplementary) that will recapitulate the features described above of the integrase mutants?

12. L297. "50mM MgSO4" is it a typo? 50mM seems very high.

13. Figure 3 legend: "(shown in red and blue)" Unless my eyes are playing tricks on me, the red is rather magenta.

14. Sup figure 1. The schematic of the distal dimers is confusing as it binds the viral DNA whereas from the structures it is more toward the target DNA.

15. Sup figure 3. The color code for the blue rectangles, red filling and red amino acids is absent.

16. Sup figure 2, 10 and 11. The elution profile of the octameric CSC has different retention volume. Please explain.

We thank the three expert reviewers for their very helpful comments and suggestions to improve the manuscript. The authors point-by-point responses are shown below in blue. Major revisions to the manuscript are italicized. We modified the Abstract to bring down the word count closer to the recommendation listed in revised manuscript submission file checklist.

Reviewer #1 (Remarks to the Author):

In this manuscript, Pandey et al determine the cryo-EM structure of the RSV cleaved synaptic complex (CSC) intasome bound to the experimental inhibitor MK-2048. They compare the structure of the CSC intasome to an X-ray structure of the strand transfer complex (STC) intasome determined previously, and through this comparison, they speculate on the molecular mechanisms underlying viral intasome assembly and function, an important area of investigation in the field. Their major claim is that the distal integrase (IN) dimers are flexible in the CSC but rigid and bound to target DNA in the STC. They conclude from these differences that the distal IN dimers rigidify upon target DNA capture and promote the formation of the target DNA-bound complex, i.e. that there is an ordered conformational transition between the key intermediates along the RSV integration pathway. While this might be the case, and is indeed one logical way of thinking about an assembly mechanism, the experimental data currently does not unambiguously support this claim. Most importantly, there is a fundamental issue with comparing the RSV CSC obtained by cryo-EM with the RSV STC obtained by X-ray crystallography. In the former, the CSC is in solution, and is therefore free to adopt a wide variety of conformational arrangements (some of which are observed in the current work). In the latter, the STC is locked into a single conformation through crystallographic packing. In most cases, I would not have an issue comparing cryo-EM to X-ray structures, for example when these authors discuss the configuration of the drug MK-2048. However, the real focus of the current work is specifically on dynamics and structural transitions, and therefore the comparison of the two structures is akin to comparing apples to oranges. Clearly, the authors understand this point (page 10, line 235), but chose not to address it prior to submission of the work. This, to me, is a sufficiently critical limitation, and therefore warrants obtaining either a cryo-EM structure of the STC or an X-ray structure of the CSC prior to making the current conclusions. For this reason, I believe that major revisions are required for this manuscript in its current form, and the authors would need to obtain such experimental data.

Authors- We thank the referee for their critical review. From communications with the manuscript Editor, we were specifically informed that new 3D structures, i.e. of STCs by cryo-EM or CSC by X-ray diffraction, would not be mandatory for the paper to move forward. The revised manuscript accordingly does not contain these requested additions. At the same time, we amended the paper to downplay the apples to oranges comparison highlighted by the reviewer. We removed from the title mention of intasome dynamics and its ability to form rigid STC in presence of target DNA. We also modified our presentation and the Discussion of the subsequent structural transitions between the CSC and the STC structures. We also removed a supplementary figure 7 in version 1 comparing the RSV CSC and STC atomic models. However, we would like to reiterate here that the 5 clusters identified by 3-dimensional variability analysis (Fig. 4) display distinct conformations than the STC crystal structure suggesting that the differences between two complexes are not simply due to two different methods used for structure determination.

The idea that RSV intasomes transition from a tetramer to an octamer assembly has been argued previously in several papers, and the authors hypothesize that a tetrameric intasome with two independent IN dimers bound to two vDNA ends is a precursor to the bona fide octameric intasome. While prior biochemical data partially support this view, there is no structural evidence for this transition, and therefore this assembly mechanism remains speculative. An alternative explanation is that the tetrameric INSTI-trapped intasome actually represents an individual “half-intasome” (with one vDNA). This might still be an on-pathway intermediate, but would not require dramatic structural rearrangements for transitioning into and forming the complete octameric RSV intasome. Please remove specific language that suggests that the authors have a thorough understanding of what the tetrameric “intasomes” represent.

We agree with the reviewer’s assertion that we do not yet have structural evidence to demonstrate the transition of tetrameric intasome to octameric intasome in the RSV concerted integration pathway. However, we should have probably reiterated in the manuscript our earlier FRET studies of both the tetrameric and octameric intasomes (Pandey et al., JBC 2017). FRET studies demonstrated the presence of two viral DNA molecules in close proximity within the RSV tetrameric as well as octameric intasomes.

Based on these studies, we respectfully disagree with the reviewer’s suggestion that tetrameric intasome is a “half-intasome” with one viral DNA molecule. Such an intasome structure would not give a FRET signal. The FRET data highly suggests the presence of two viral DNA molecules in the tetrameric RSV CSC prior to its conversion to the octameric form.

Furthermore, our previously published X-ray crystallography structural studies (Shi et al., PLoS One 2013; Yin et al., Nature 2016) demonstrated that binding of viral DNA to a proximal dimer does not introduce significant structural change. *i.e.* the CCD–CTD configuration for the viral DNA bound proximal IN dimers is similar to DNA-free RSV IN. However, binding of the two distal IN dimers requires significant conformation change specially in CCD-CTD linker and CTD to adeptly bind to the two proximal dimers bound with viral DNA.

It is possible that the precursor to tetrameric intasome could be an intermediate consisting of an IN dimer bound to one viral DNA.

In summary, we did not mean to exclude other models for the tetrameric CSC. We have modified this paragraph in the Discussion to address the reviewer’s comments. Modifications are in blue.

Page 11 (Lines 258-267)

“The precursor to the RSV tetrameric CSC is unknown. However, FRET analysis of both the tetrameric and octameric CSC intasomes demonstrated that both contain two viral DNA molecules with their 5’-ends in close proximity (15).” Trans-communication can occur between wt IN dimers bound to two different 3’ OH recessed viral DNA ends *i.e.* an IN dimer bound to GU3 can couple with another IN dimer bound to a defective U3 end that by itself is incapable of assembling the CSC (16). *“We speculate that the assembly process occurs in a stepwise fashion, possibly by formation of the tetrameric intasome from two independent proximal IN dimers bound to viral DNA ends, followed by binding of the flexible distal dimers donating the critical CTDs into the CIC of the mature octameric CSC.”*

Additional minor comments are below:

Figure 1. Why is the model shown before the map?

Authors- We feel the model provides a clearer interpretation of the RSV CSC cryo-EM structure and function. From the literature, we note that maps and models are invariably presented in either order.

Figure 3. This figure needs to be improved. These are raw snapshots from Chimera, and even the colors haven't been changed from their defaults. Some way of synthesizing this information would go a long way.

Authors- The 3-dimensional variability analysis was performed at 7Å resolution to observe the large variability among particles having conformational heterogeneity. We modified the figure to improve the image resolution and background. We feel this is the optimum way to represent this type of analysis along with the Supplemental movie. We extended the 3-dimensional variability analysis by re-extracting the particles at full resolution from each cluster and refining their structures by homogeneous refinement. We have shown the model obtained from one cluster having best resolution in the new Supplementary figure 9.

Supplementary Figure 1. "present" should be "presence"

Correction was made.

Supplementary Figure 2. Please show a gel as well.

Authors- We are unsure from reviewer comment what kind of gel is requested? We show a gel demonstrating the integration activity profile associated with the octameric intasome (Supplementary figure 10). We have not been successful with electrophoresis of assembled RSV IN/DNA complexes into native agarose gels presumably due to their overall high positive charge. We have performed SEC-MALS on both the tetrameric ($151,000 \pm 2000$ Da) and octameric CSC ($257,000 \pm 8,000$ Da) stabilized by MK-2048 establishing their molar mass (Pandey et al., JBC 2014, JBC 2017). The molar mass of the RSV STC was $252,000 \pm 9,000$ Da (Yin et al., Nature 2016). The tetrameric CSC was produced by IN 1-269 and the octameric CSC and STC by IN 1-278.

Supplementary Figure 4. The half-map and map-model FSC curves should be shown on the same scale for easier comparison. It looks like the model only correlates to the map up to a resolution of $\sim 1/0.26$ (~ 3.8 Å), whereas the map is resolved to ~ 3.2 Å. Ideally, these should be very similar. This discrepancy should be addressed. Either (1) the model would need to be improved, or (2) the FSC curves need to be obtained and compared for the most homogeneous regions of the map (presumably CIC). Ideally, one would also show 3D FSCs.

Authors- As per the reviewer suggestion, we have shown the map-map FSC and model vs map FSC plot on same scale (Supplementary figures 5G and 5H). To address the discrepancy of model having lower resolution than map, we obtained the FSC plot for the most homogeneous

region (CIC) only and it improved the model vs map FSC (Supplementary figure 5H). We thank the reviewer for this suggestion. We have also included 3D FSC for the map (Supplementary Fig 5G).

Supplementary Figure 5. How was local resolution calculated? It looks broken up and not very smooth, as one would expect.

Authors- As mentioned in the supplementary figure legend and the method section, the local resolution was determined in cryoSPARC and surface color displayed in Chimera. A smoother map for the CIC region is now provided in Supplementary figure 5I.

Supplementary Figure 10. How does the activity of the tetrameric CSC or IN alone compare with that of the octameric CSC in concerted integration?

Authors- We cannot compare the concerted integration activity of the tetrameric or octameric CSC because of the presence of MK-2048 in these intasomes. Only the octameric CSC can be isolated without MK-2048 present (Supplementary figure 10). Tetrameric CSC produced by IN 1-269 in the absence of MK-2048 cannot be isolated by SEC (Pandey, et al., JBC 2014).

We have compared the concerted integration of C-terminal truncated IN 1-269, IN 1-274, IN 1-278 and wt IN 1-286 as well as their 3' OH processing activities (Pandey, et al., JBC, 2017) (Shi, et al., PLoS One, 2013). They are essentially equivalent for both concerted integration and 3' OH processing activities at 37°C.

Supplementary Figure 13. This is negative stain or cryo? If negative stain (as in the legend), why is the contrast reversed? Why is this figure last? If negative stain, presumably this should be shown before the cryo-EM data.

Authors- The micrograph shown in the Supplementary figure 13 was indeed from a negative stain. During the data analysis in cryoSPARC, the image was contrast reversed. We have now provided the usual looking negative stain micrograph image (white particles in black background) in Supplementary figure 4. We have now moved this supplementary figure before the cryo-EM data as per the suggestion of the reviewer.

Table S1.

- Why was no symmetry imposed?
- Clashscore is quite high. Please address.

Authors- We had performed the reconstruction with C2 symmetry and it did not improve the resolution. To reduce the clashscore, the model was further refined and it reduced the clashscore from 26.42 to 16.42.

Reviewer #2 (Remarks to the Author):

This manuscript reports a cryo-EM structure of the RSV intasome prior to its engagement with target DNA. As such, the new data nicely complement and extend prior work by the same group that described the strand transfer complex (i.e. intasome plus target DNA, published in 2016).

The current structure was determined in the presence of a strand transfer inhibitor MK2048. The structure plus mutational data seem to explain why some of these small molecules inhibit RSV replication (RAL, MK2048, etc), while some (EVG) do not. This part of the story is interesting, straightforward and arguably important.

The authors then decide to delve into the mystery of the C-terminal tail of RSV integrase. They have published at least two papers on this tail, and frankly the current work does not add any clarity. They do show that the tail is not essential for RSV replication, and as far as I understand, they do not see it in their reconstruction. I feel that too much emphasis is made on that tail, which distracts from the key findings. Annoyingly, it is difficult to understand this part of the manuscript without downloading the previous papers about the tail. In addition to adding clarity, it would help to include a panel (within a main figure) showing the sequence of the tail, indicating which tail residues are visible in the current (and previous?) structure.

Authors- We should have better explained why we wanted to investigate the connection between the CTD and the “tail” region of RSV IN to assemble the precursor tetrameric intasome on route to the mature octameric structure. We modified this section of the manuscript accordingly as described below

Page 8 (Lines 170-180)

Functional CTDs and “tail” region boundaries for CSC assembly and RSV infectivity. We next wanted to understand the interactions between the CTD and the “tail” region of IN to promote assembly of intasomes and to foster virus infectivity. CTDs play critical roles in stabilizing functional retrovirus intasome CICs (1,3,4,6,8). The RSV IN “tail” region (defined as residues from Ile269 to Ala286; Supplementary Fig. 3) accelerate the conversion of the precursor tetrameric CSC to the mature octameric form in a time and temperature-dependent manner (15,16). RSV IN 1-269 catalyzes 3' OH processing of blunt-ended viral DNA at wt IN levels and concerted integration into supercoiled target DNA at ~70% level of wt IN (14). IN 1-269 tetrameric CSCs could be isolated only in the presence of INSTI only (14) while IN 1-278 octameric CSCs could be isolated in the presence (Supplementary Fig. 2) or absence of INSTIs (Supplementary Fig. 10).

We addressed the “tail” sequence questions as follows. The sequence of the RSV IN tail region is shown in Supplemental figure 3 and now also in Fig. 5. In our previous structural studies of IN (1-270), the C-terminal 16 aa of IN (tail region) was removed thus allowing IN to be crystallized (Shi, et al., PLoS One, 2013) and in the STC intasome (Yin, et al., Nature, 2016). Earlier crystallographic studies of RSV two domain IN construct (49-286) possessing this entire tail region demonstrated it was disordered (Yang, et al., JMB, 2000).

In the current octameric cryo-EM intasome, which was produced using IN (1-278), the last ordered residue was Ile269 on all eight chains. We added a sentence in the revised manuscript to clarify this point. (Page 4, Line 94)

We also modified Fig. 5 to add clarity showing the sequence of the C-terminal residues of the RSV IN tail (truncations marked in red) for comparison to similar C-terminal truncations of HIV-1 by other labs. The atomic structure of the HIV-1 tail region (Supplemental figure 3) has not been resolved (Chen, et al., PNAS, 2000; Eijkelenboom, et al., Nat. Struct., 1995, Passo et al., Science 2017, Passo et al., Science 2020).

Further points:

By the look of it, RSV intasome appears very similar to the cryo-EM reconstruction of the MMTV intasome (EMD-6441). It would be good to compare or at least mention.

Authors-We had mentioned in the Discussion that the MMTV octameric intasome possessed flexible distal dimers and the critical role these CTD dimers are in the formation of the CIC and thus the stability of the intasome. (L228 to L231 in version 1, Lines 255-257 in the revised version)

Line 355: "homologous refinement"

The particle number in the final set is very large. Fig S4 does not show results of a 3D classification; was it even done (beyond ab-initio)? Relion is much more robust in 3D classification than heterogenous refinement in Cryosparc, for example; and ab-initio is not very powerful as a 3D classification tool. My impression is that cryo-EM data processing was done without due care.

Authors- We are sorry that method section was not sufficiently clear. We had described the cryo-EM imaging analysis workflow in method section. (Line 352-355 in version 1). Particles from selected 2D classes were used for ab-initio reconstruction in 5 classes. All of these classes were homogeneously refined and the best class with highest resolution and coverage (supplementary figure 4E in version 1 and now supplementary figure 5E) was selected for further analysis and model building. We have expanded the method section to explain it in more detail in the revised manuscript (Line 352-375).

We had tried heterogeneous as well as non-uniform refinement methods and it did not yield improvement in the resolution. As we mentioned on Line 360-361 (first version), we had used RELION to analyze half of our dataset and it provided similar results. In our hands, cryoSPARC allowed faster data processing using less computing hours and provided better resolved maps. Overall, we think we have taken appropriate care and due diligence in all of the experiments including cryo-EM data analysis.

I would expect non-uniform refinement in Cryosparc to work better in this case, where considerable disorder (distal dimers) is present (this might not be the case, but have the authors attempted it?).

Authors- We had performed non-uniform refinement and it did not improve the resolution.

According to Table 1, 2-fold symmetry was not used during reconstruction. Was there a particular benefit from not imposing C2?

Authors- We had performed the reconstruction with C2 symmetry and it did not improve the resolution.

The micrograph in Fig S13A does not look like negative stain!

Authors- The micrograph shown in the Supplementary figure 13 was indeed from a negative stain. During the data analysis in cryoSPARC, the image was contrast reversed. We have now

provided the usual looking negative stain micrograph image (white particles in black background) in Supplementary figure 4. As suggested by reviewer 1, we have now moved this figure earlier before the cryo-EM data.

Reviewer #3 (Remarks to the Author):

The manuscript Pandey et al. is revolving around the structure of the Rous sarcoma virus (RSV) octameric intasome intermediate called cleaved synaptic complex (CSC) and the role of the integrase C-terminal tail region residues in the infectivity and intasome assembly.

Compared with the already published RSV intasome strand transfer complex (STC) structure, the CSC shows an unexpected flexibility of its distal protomers. The functional consequence of this dynamic flexibility is however not clear and remains an interesting question to further address.

As the assembly procedure requires the stabilization with an integrase strand transfer inhibitor (INSTI), the author could also explain the molecular mechanisms involved in RSV Elvitegravir (EVG) resistance.

Finally, using a pseudotyped RSV reporter system, the author showed that the last 17 residues “tail region” were dispensable for RSV infection, data consistent with their earlier report using in vitro integration assay (Bera et al. 2018).

I don't have any major comments on the quality of the data themselves. This is a beautiful intasome structure revealing an unexpected feature of the distal dimers.

However, I have some questions and comments that might need to be addressed before publication.

1. L119-120. Is there a technical or biological explanation as to why one distal dimer shows minimal density?

Authors- Overall, there was lower electron density in both of the distal dimers. The CIC region containing two proximal dimers bound to DNA and distal CTD contain several inter-subunit interactions, thereby providing stability and higher electron density. However, the distal CCD-NTD regions do not seem to have multiple interactions with the CIC region and hence very flexible leading to conformational heterogeneity. We believe this flexibility could be main reason why NTD-CCD of distal dimers show minimal density. In our previously published RSV STC structure, we showed that the distal NTD-CCDs were stabilized by interacting with target DNA (Yin et al., Nature 2016). In cryo-EM imaging, it could be simple lower mathematical probability of having both dimers (NTD-CCD) in perfect orientation. We did not impose any symmetry during the refinement, hence the density is weaker on one side and reflect the true orientation of particles during imaging.

Similar conformational and compositional heterogeneity with flanking subunits has been observed in several of retroviral intasomes structures published *i.e.* for MMTV (Ballandras-Colas et al., Nature, 2016); HIV-1 (Passo et al., Science, 2017; Passo et al., Science 2020; Li et al., JMB, 2020) and SIV (Cook et. al., Science, 2020).

2. L185. Considering the apparent important role of the tail region in the kinetics of intasome assembly I felt a bit disappointed by the sole infection experiment. One could not rule out that

although the relative infectivity is similar 43h post transduction, the kinetics/timing of integration might be different between the truncated IN mutants. This can be addressed by synchronizing the infection and adding RAL or DTG to the cells at different time points (See similar experiment: Zurnic et al. PLoS Pathog 2016).

Authors- As a point of clarity, the infection time course was 48h (5h of virus absorption, which was followed by 43h incubation after virus wash off); we amended this sentence in Methods for clarity.

Although we do not disagree that expanding the virology section of the paper considerably might bring out subtle differences between some of the infectious RSV IN tail mutant viruses, we would argue that such work exceeds the scope of the paper because it would not add significantly to the main conclusion reached by performing the infections under standard operating conditions, i.e. 2 days following transduction with VSV-G pseudotyped single-round reporter virus. The main point of this section of Results was to test whether the tail region of IN was required or was dispensable for RSV infection. The data clearly show that the IN tail is dispensable for RSV infection under these standardized conditions.

To further clarify this aspect of the paper, we have added an HIV-1/RSV tail region sequence alignment to main Fig. 5 that also summarizes results of prior HIV-1 deletion mutagenesis and infection measures. We accordingly expanded this section of Results to clarify the strategy for performing the tail region mutagenesis experiments in the context of infectious virus and also how these findings impact our interpretation of RSV intasome functionality; for more details, please see the response to Reviewer 2 comments.

These truncation mutants might also show a different integration site selection profile. This can be discussed.

Authors- We suppose it may be theoretically possible that the truncation of the tail region could affect the integration site selection for RSV; we suspect the reviewer's comment may be rooted in the knowledge that the tail region of MLV IN interacts with cellular BET proteins to guide MLV integration to promoter regions. However, to the best of our knowledge, a similar cofactor of RSV IN that might determine the genomic pattern of RSV integration has not been identified in the literature. Moreover, if such a cofactor even exists, that it will interact with the RSV IN tail region is not known. Accordingly, we opted to avoid discussing this in the paper.

3. Paragraph starting at L187. To be honest, I don't see the rationale behind this mutational analysis (S262/R263) to understand the function of the tail region during infection. Can you explain?

Authors-Thank you for this comment. S262/R263 are not the part of tail region, rather they are in CTD region. In response to reviewer 2 comments, we provided explanation to the reasoning behind modifying the residues in CTD and tail region of IN. Please see below the modified section in the manuscript-

Page 8 (Lines 170-176)

Functional CTDs and "tail" region boundaries for CSC assembly and RSV infectivity. We next wanted to understand the interactions between the CTD and the "tail" region of IN to

promote assembly of intasomes and to foster virus infectivity. CTDs play critical roles in stabilizing functional retrovirus intasome CICs (1,3,4,6,8). The RSV IN “tail” region (defined as residues from Ile269 to Ala286; Supplementary Fig. 3) accelerate the conversion of the precursor tetrameric CSC to the mature octameric form in a time and temperature-dependent manner (15,16).

Page 9 (Lines 205-216)

Due to the importance of β 10 for RSV/HIV-1 infection, we targeted both conserved and variable residues that comprised the η 3 helix that lies immediately upstream from β 10. IN mutant viruses R263A and R263K, which altered conserved RSV IN residue Arg263, supported approximately 20% and 50% of the level of wt 1-323 infection, respectively (Fig. 5C). Both purified RSV IN mutants were unable to form the octameric CSC while IN R263K was only capable of producing the tetrameric CSC (16) suggesting Arg263 in the distal IN dimer may play an important role in octameric intasome assembly (Supplementary Fig. 6B). While η 3 helix IN mutant virus S262P that targeted variable RSV IN residue Ser262 was noninfectious, S262R displayed ~25% of wt infectivity (Fig. 5B), similar to what was previously observed under conditions of spreading RSV replication (30). Recombinant IN S262P was defective for CSC assembly while S262R formed octameric CSC intasomes, albeit at lower efficiency than wt IN (Supplementary Fig. 11).

In brief, the mutational analysis of S262/R263 were performed in the critical η 3 helix between β 9 and β 10 (see Supplementary figure 3) in the CTD, adjacent to the tail region. These mutations established the critical role of η 3 in the CTD for assembly of the CSC and infectivity. Both S262 and R263 appear to play critical roles in the stability of the octameric CSC (Supplementary figures 6 and 11) and the assembly of the octameric CSC (Supplementary figure 11)(reference 16 in manuscript).

4. L231-234. I think the comparison as such is fair but we could argue that i) PFV intasome is fully tetrameric and do not require/have distal protomers outside the conserved intasome core and ii) MVV intasome was assembled using LEDGF/p75 which is a priori crucial for complex formation and might prevent further flexibility. For these reason I think it would be important to mention the differences between the models.

Authors- We agree that PFV intasome is less flexible because it only has the tetrameric core. We mention in the preceding sentence that PFV and MVV intasomes are tetrameric and hexadecameric, respectively. We also included a reference to how the reduced flexibility of MVV intasome may have relevance to binding of the lentivirus-specific cofactor LEDGF/p75. (Lines 243-249)

5. L263-267. The extreme tail region of gammaretroviral integrase CTD is involved in the interaction with the cofactors Brd2-4. The interaction stimulates MLV in vitro integration activity. Moreover, ALV integrase binds to the FACT complex via its CTD, interaction that also stimulate in vitro integration. What is the status of RSV IN with regards to FACT? Is FACT a cofactor for alpharetroviral integrases or only ALV?

Authors- The RSV and ALV INs have the same amino acid sequence. Based on this, one would assume that FACT may also bind RSV IN. We are unaware of any studies on the role of FACT in alpharetrovirus infection or on the interaction of FACT with alpharetrovirus IN beyond the sole

paper on ALV from the Beemon laboratory that the reviewer has referred to in their comment. As alluded to above, there is no data in the Beemon paper as to whether FACT influences sites of alpharetroviral integration in infected cells.

Depending on the answer, biochemical analyses with the mutants/truncations including FACT might reveal some interesting features.

Authors-Although we do not disagree with the reviewer's comment that studies with the FACT complex could be of potential interest, we feel this work exceeds the scope of this paper. First off, recombinant FACT proteins (SSRP1 + SUPT16H) would need to be expressed and purified as recombinant protein reagents, which would be a new direction for our laboratories. Second, biochemical measures would need to be done to verify a presumed interaction with RSV IN and potential stimulation of wt and RSV IN mutant concerted integration activities in vitro. Third, genomic integration site targeting measures would need to be done with infected cells to establish whether the presumed RSV-IN interaction in any way influences sites of RSV integration into cellular chromosomes. Since we have yet to report on alpharetrovirus integration sites, these LM-PCR and Illumina sequencing pipelines would need to be established. Such work clearly culminates as a stand-alone study. We would remind the reviewer that the virus measurements we have performed (Fig. 5) show clearly that the IN tail region is dispensable for RSV infection in cells that almost certainly express the FACT complex.

Also, infections were done in DF-1 cells, do you see a different infection phenotype using a heterologous mammalian system to change the cellular context?

Authors- We would argue that the most physiological cell types for infection assays are those derived from the same animal species as where the virus has persisted and evolved for millennia. Since DF-1 are chicken cells, we do not see the point of analyzing heterologous cells such as those derived from mammalian species.

Minor concerns:

6. L53. spumavirus is the old taxonomy, since 2019 the PFV genus is "Simiispumavirus"

Authors- We have corrected the genus as suggested and thank the reviewer for suggesting the correction.

7. L61. "RSV IN is homodimeric". Is it meant "in solution"?

Authors- RSV IN is homodimeric is "in solution". This was added to the revised manuscript.

8. L92. "determined at 3.8A" add "resolution"

Authors- "resolution" was added.

9. Sup figure 8 and 9. Some of the spheres and details are hard to see. Increasing the surface transparency might help.

Authors- We have adjusted the transparency in Supplementary figures 7 and 8. Thank you for the suggestion.

10. L224. Typo: confirmation instead of conformation.

Authors- "conformation" was inserted.

11. I found a bit difficult to keep track with the IN truncations and the point mutants in regards to their propensity to forms tetrameric vs octameric CSC vs STC, their in vitro activity and infectivity. Is it possible to create a table (as a figure or supplementary) that will recapitulate the features described above of the integrase mutants?

Authors-We did not present any data on STC formation in this manuscript. From our previous publications, all C-terminal IN truncations (1-269 to full-length IN 1-286) shown in Fig. 5A and 5B assemble the STC. In addition, single-point IN mutation R263A and R263K produced the STC but were only able to produce the tetrameric CSC. The concerted integration activities of these IN mutants were ~20 and 50 % of wt IN, respectively, matching their relative infectivity (Fig. 5C).

We did not investigate whether single-point IN mutants S262P and S262R assembled the STC but only the assembly of the CSC and their integration activities (Supplementary figure 11).

We summarized this above information in the last four sentences of the second paragraph in the revised manuscript. (page 9, Lines 207-216)

12. L297. "50mM MgSO₄" is it a typo? 50mM seems very high.

Authors- The concentration is 50 mM.

13. Figure 3 legend: "(shown in red and blue)" Unless my eyes are playing tricks on me, the red is rather magenta.

Authors- We changed the word red to "magenta" in the figure legend.

14. Sup figure 1. The schematic of the distal dimers is confusing as it binds the viral DNA whereas from the structures it is more toward the target DNA.

Authors- This is a cartoon only depicting a simplistic representation of CSC assembly. We have modified the figure to more accurately reflect the model obtained from the cryo-EM structure.

15. Sup figure 3. The color code for the blue rectangles, red filling and red amino acids is absent.

Authors- The following statement was inserted to the legend of Supplementary figure 3. "The RSV IN sequence (205-286 aa) was aligned in ClustalW and the alignment was used to generate the figure in ESPript3.0. The coloring scheme followed standard ESPript standards (Robert, X. and Gouet, P. (2014) "Deciphering key features in protein structures with the new ENDscript server". Nucl. Acids Res. 42(W1), W320-W324.)

16. Sup figure 2, 10 and 11. The elution profile of the octameric CSC has different retention volume. Please explain.

Authors- The different elution volumes for octameric is due to Supdex-200 Increase column bed height adjustment over the period of usage. We have confirmed the column efficiency and elution volumes by comparing with molecular weight standards for size-exclusion chromatography at all times. All three elution profiles included size standards.

Reviewers' comments:

Reviewer #1 (Remarks to the Author):

This manuscript presents the novel structure of the RSV cleaved synaptic complex intasome. The authors have improved the manuscript from the original submission, and I would like to thank them for acknowledging and responding to the reviewer comments. With respect to the comparison between X-ray and cryo-EM structures, if the editor is in agreement that no further structural biology data is necessary for publication, then I will not contest with the decision. However, I maintain that, since the motivation of the work pertains to intasome assembly and dynamics, a comparison between a single static X-ray structure that is restrained by crystal lattice packing and a broader set of cryo-EM structures that are (effectively) in solution, is not fully justified. For example, I disagree with the following statement in the discussion. "However, all 5 representative conformations of the CSC in the 3DVA are distinct from that observed for the RSV STC structure, suggesting that their differences are not simply due to the different structural biology methodologies." The differences may precisely be due to the structural biology methodology, if, for example, x-ray lattice constraints would lock into place distal IN dimers connected by NTD:CCD linkers that would, under normal circumstances, be flexible and dynamic. Therefore, I would suggest to remove all statements from the manuscript suggesting that the authors have an understanding of the mechanism of assembly that stems from these distinct structures.

Supplemental data.

Figure S5G. Legend should read "Histogram of 1D FSC values overlaid with the average global FSC curve, shown alongside the binarized 3D FSC volume". At what threshold is the 3D FSC volume displayed? In practice, the 0.5 threshold provides a good visual representation of the 3D FSC anisotropy (although the 0.143 threshold is, technically, the correct threshold to use).

Figure S5H. The nominal resolution value for the map/model FSC is always taken at 0.5, not 0.143, as is done in the revised version of the manuscript. There is still a discrepancy between the experimental half map resolution of 3.2 Å and the map/model resolution, which I am eyeballing at $\sim 1/0.27$ (~ 3.7 Å). The same comment is made as in my original review, and the authors still need to address this point by either (1) improving the model or (2) obtaining the plots for the most homogeneous (modeled) regions. I would suggest to try using the Phenix plots, which may better accomplish the latter.

C2 symmetry should improve the CIC, but not necessarily the global resolution. I don't understand why the resolution was not improved after applying 2-fold symmetry. A rationale or justification should be provided in the final manuscript.

I generally have concerns that the variability analysis is over-interpreted, see below:

I interpret the varying density for the distal NTD:CCD dimers rather simply – each dimer is connected by flexible linkers, and therefore each one is independently flexible about the CIC. Subdividing the data into 5 states is not expected to be sufficient to capture the full extent of flexibility under such conditions. The algorithm for variability analysis would hone in on one dimer, whereas the other would occupy some distinct position. This is why, during asymmetric refinement, one of the distal dimers is always poorly resolved.

"The outer subunits in the proximal dimer have a "collapsed" conformation demonstrating an outward concerted movement of at least 5 Å when the distal NTD-CCD is further away from the CIC". – Based

on what evidence?

"When the distal NTD-CCD is closer to the proximal dimers, it makes the proximal dimers more compact". – Based on what evidence?

What is "reaction coordinate space"?

Reviewer #2 (Remarks to the Author):

Overall, the manuscript improved in revision.

One final, very minor comment:

Abstract could do with proofreading ("Unexpectedly, the last 17 amino acids of IN were dispensable for virus infection but regulates..").

Reviewer #3 (Remarks to the Author):

Happy with the revisions.

** See the Nature Portfolio author and referees' website at www.nature.com/authors for information about policies, services and author benefits

We thank the Editor and reviewers for comments to improve the revised manuscript. We have addressed the comments put forwarded by reviewer #1 in blue.

Reviewers' comments:

Reviewer #1 (Remarks to the Author):

This manuscript presents the novel structure of the RSV cleaved synaptic complex intasome. The authors have improved the manuscript from the original submission, and I would like to thank them for acknowledging and responding to the reviewer comments. With respect to the comparison between X-ray and cryo-EM structures, if the editor is in agreement that no further structural biology data is necessary for publication, then I will not contest with the decision. However, I maintain that, since the motivation of the work pertains to intasome assembly and dynamics, a comparison between a single static X-ray structure that is restrained by crystal lattice packing and a broader set of cryo-EM structures that are (effectively) in solution, is not fully justified. For example, I disagree with the following statement in the discussion. "However, all 5 representative conformations of the CSC in the 3DVA are distinct from that observed for the RSV STC structure, suggesting that their differences are not simply due to the different structural biology methodologies." The differences may precisely be due to the structural biology methodology, if, for example, x-ray lattice constraints would lock into place distal IN dimers connected by NTD:CCD linkers that would, under normal circumstances, be flexible and dynamic. Therefore, I would suggest to remove all statements from the manuscript suggesting that the authors have an understanding of the mechanism of assembly that stems from these distinct structures.

We have removed statements from the manuscript regarding the comparison of the cryo-EM CSC structures to the x-ray structure of the STC with respect to assembly mechanisms. We discuss the flexibility in the cryo-EM CSC structure by itself. We at the same time feel that it is unlikely that the crystal lattice contacts observed in the x-ray structure necessarily yielded an STC conformation that does not exist in solution. For clarity, we have revised the Discussion as follows:

"Further structural work, for example via solving the RSV STC structure by cryo-EM, will be needed to ascertain how the CSC converts to STC upon target DNA binding and how target DNA stabilizes the flexible NTD-CCDs in distal subunits". Page 10.

Supplemental data.

Figure S5G. Legend should read "Histogram of 1D FSC values overlaid with the average global FSC curve, shown alongside the binarized 3D FSC volume". At what threshold is the 3D FSC volume displayed? In practice, the 0.5 threshold provides a good visual representation of the 3D FSC anisotropy (although the 0.143 threshold is, technically, the correct threshold to use).

The figure legend is modified as suggested including the threshold value. The 3D FSC volume was displayed at 0.143 threshold.

Modified legend for Fig. S5G reads as below-

“Histogram of 1D FSC values overlaid with the average global FSC curve, shown alongside the binarized 3D FSC volume displayed at 0.143 threshold”.

We also modified the legend for Supplementary Fig. 9C to maintain uniform language.

Figure S5H. The nominal resolution value for the map/model FSC is always taken at 0.5, not 0.143, as is done in the revised version of the manuscript. There is still a discrepancy between the experimental half map resolution of 3.2 Å and the map/model resolution, which I am eyeballing at $\sim 1/0.27$ (~ 3.7 Å). The same comment is made as in my original review, and the authors still need to address this point by either (1) improving the model or (2) obtaining the plots for the most homogeneous (modeled) regions. I would suggest to try using the Phenix plots, which may better accomplish the latter.

We have added the nominal resolution value at FSC cutoff 0.5. It was indeed 3.7 Å as the reviewer inferred. We think the map to model resolution is lower because the map has a variable resolution. The variability impinges on the ability for the model to explain the map density as well as might be expected. Similar variability is described in literature (Rosenthal PB and Rubinstein JL 2015, Validating maps from single particle electron cryomicroscopy. *Current opinion in structural biology*, 34:135-44).

We also modified Supplementary Fig. 9D to maintain uniformity.

The plot data shown in Supplementary Fig. 5H was derived from Phenix.

C2 symmetry should improve the CIC, but not necessarily the global resolution. I don't understand why the resolution was not improved after applying 2-fold symmetry. A rationale or justification should be provided in the final manuscript.

We agree that C2 symmetry should have improved the CIC resolution. However, in our dataset, we did not observe much improvement in resolution upon imposing C2 symmetry. It's possible that the flexible nature of CSC and CIC could have minute differences between parts of the complex as discussed in Huiskonen JT review article (*Bioscience reports* 2018, Image processing for cryogenic transmission electron microscopy of symmetry-mismatched complexes, 38:BSR20170203) and Serna M (*Frontiers in Molecular Biosciences* 2019, Hands on methods for high resolution cryo-electron microscopy structures of heterogeneous macromolecular complexes 6:33). We have added a sentence in manuscript as suggested by the reviewer.

The modified section reads as below;

“The 3D reconstructions were refined by homogeneous refinement without imposing symmetry. Using C2 symmetry during the refinement did not significantly improve the map resolution possibly due to the flexible nature of the complex”. Page 15.

I generally have concerns that the variability analysis is over-interpreted, see below:

I interpret the varying density for the distal NTD:CCD dimers rather simply – each dimer is connected by flexible linkers, and therefore each one is independently flexible about the CIC. Subdividing the data into 5 states is not expected to be sufficient to capture the full extent of flexibility under such conditions. The algorithm for variability analysis would hone in on one dimer, whereas the other would occupy some distinct position. This is why, during asymmetric refinement, one of the distal dimers is always poorly resolved.

The major objective of three-dimensional variability analysis (3DVA) was to demonstrate the conformational heterogeneity. It showed a continuous family of 3D structures for the CSC particles specifically with regards to the flexible distal subunits due to their linkers (Fig. 4G and Supplementary movie).

Cluster analysis was a secondary result of 3DVA in determining the types of discrete conformations within the continuous family of 3D structures of the CSC. Hence, cluster analysis was not used as primary analysis to demonstrate the “full extent of flexibility”.

We agree with the later part of reviewer’s explanation for why one of the distal dimers is always poorly resolved. We provided a similar answer to Reviewer #3 in the first revision of the manuscript.

“The outer subunits in the proximal dimer have a “collapsed” conformation demonstrating an outward concerted movement of at least 5 Å when the distal NTD-CCD is further away from the CIC”. – Based on what evidence?

It was based on the Supplementary movie and the measurements done in Chimera at different frames.

We changed the word “collapsed” to “relaxed” as it more correctly reflects the structure (Page 7).

“When the distal NTD-CCD is closer to the proximal dimers, it makes the proximal dimers more compact”. – Based on what evidence?

It was based on the Supplementary movie and the measurements done in Chimera at different frames.

What is “reaction coordinate space”?

For a detailed mathematical explanation of how this term is derived, we would refer to the preprint citation by Ali Punjani and David Fleet (ref 23). In general terms- reaction coordinate space is simply the position of each particle along each mode in 3D space.

Reviewer #2 (Remarks to the Author):

Overall, the manuscript improved in revision.

One final, very minor comment:

Abstract could do with proofreading ("Unexpectedly, the last 17 amino acids of IN were dispensable for virus infection but regulates..").

We made the correction and modified the sentence.

Reviewer #3 (Remarks to the Author):

Happy with the revisions.